The effectiveness of arbuscular mycorrhizal fungal species (Funneliformis mosseae, Rhizophagus intraradices, and Claroideoglomus etunicatum) in the biocontrol of root and crown rot pathogens, Fusarium solani and Fusarium mixture in pepper

Bilgili Ayşin aysin.bilgili@tarimorman.gov.tr
Plant Health Department, GAP Agricultural Research Institute , Şanlıurfa , Turkey
Yasin Nasim
Electronic publication date: 2025 Jan 16
Publication date: 2025
Volume: 13
Electronic Location ID: e18438
Received 2024 Jun 25; Accepted 2024 Oct 10
Copyright: © 2025 Bilgili
Copyright year: 2025
Copyright holder: Bilgili
License: This is an open access article distributed under the terms of the Creative Commons Attribution License, which permits unrestricted use, distribution, reproduction and adaptation in any medium and for any purpose provided that it is properly attributed. For attribution, the original author(s), title, publication source (PeerJ) and either DOI or URL of the article must be cited.
License URL: https://creativecommons.org/licenses/by/4.0/

Keywords: Pepper, AMF, Disease severity, Fusarium solani, Fusarium mix, Root colonization, Plant nutrients, Biological control, Root and crown rot

Funding: The Ministry of Agriculture & Forestry General Directorate of Agricultural Research and Policies (TAGEM) GAP Agricultural Research Institute (GAPTAEM) The Ministry of Agriculture & Forestry, General Directorate of Agricultural Research and Policies (TAGEM), and GAP Agricultural Research Institute (GAPTAEM) provided the laboratory, climate room infrastructure, and financial assistance to carry out this study. The funders had no role in study design, data collection and analysis, decision to publish, or preparation of the manuscript.

==============================
This study evaluated the effectiveness of arbuscular mycorrhizal fungi (AMF) species, including Funneliformis mosseae (FM), Rhizophagus intraradices (RI), Claroideoglomus etunicatum (CE), and a Mycorrhizal mix (MM) comprising these three species, on pepper plants (Capsicum annuum L.) inoculated with two isolates of Fusarium solani (48-F. solani and 18-F. solani) and two isolates of Fusarium mix (50-F. mixture and 147-F. mixture). Analysis of variance (ANOVA)-Tukey statistics revealed that the effects of AMF inoculations on morphological parameters, disease severity, root colonization, and total spore numbers in pathogen-infected plants varied significantly depending on the AMF species and pathogen group. AMF colonization significantly reduced disease severity, with disease inhibition (DI) reaching up to 58%, depending on the specific pathogen. However, there were a few instances where the application of AMF did not lead to a reduction in disease severity. Single AMF species were more effective in enhancing the growth of pathogen-treated host plants and suppressing disease compared to the mixed AMF. The mixed AMF was only more effective in balancing pathogen-induced decreases in plant nutrients (Copper (Cu), Magnesium (Mg), Zinc (Zn), and Phosphorus (P)). Among the compared mycorrhizae, C. etunicatum (CE) was the most effective in disease suppression due to its relatively more positive effects on plant root structure, increasing root fresh weight by up to 49% in the CE+pathogen plant group compared to the control group. Root colonization rates were generally higher in plants treated with both mycorrhiza and pathogens compared to plants treated with mycorrhiza alone. Overall, the curative effects of AMFs on plants following pathogen application varied concurrently with disease severity rates caused primarily by pathogens. AMFs demonstrated greater efficacy in combating 18-F. solani, which causes less severe plant disease. However, the effectiveness of AMFs was comparatively lower against 48-F. solani and 147-F. mix., which cause more severe plant disease. This indicates that the efficacy of AMFs varies depending on the specific strain of Fusarium solani, with better results observed against strains that cause less severe plant disease.

Introduction

The pepper (Capsicum annuum L.) plant is a globally significant and widely used crop, consumed both fresh and dried. It is cultivated over an extensive area of 2.05 million hectares, with a global annual production of approximately 35.6 million tons (Faostat, 2021). Turkey contributes around 6–9% of this total production. Pepper cultivation is often impacted by two prevalent fungal diseases: root rot and wilt. These diseases pose a significant threat to the health and productivity of pepper plants, leading to adverse effects on overall crop yield (El-kazzaz et al., 2022). Fungal diseases cause annual yield losses of approximately 14% in pepper crops (Coskun, Alptekin & Demir, 2023). Root and crown rot are common pathogens that occur during the growth period of pepper plants in the field. Over time, affected plants start to exhibit symptoms such as wilting, yellowing, and drying. The pathogen responsible for these diseases can persist in the soil for an extended duration, often in the form of chlamydospores. This soil-borne pathogen has the ability to survive in plant residues, leading to the occurrence of wilting in newly planted crops during subsequent favorable conditions for the disease (Zhang, Yu & Wang, 2021; Devi et al., 2022). The impact of wilt is not limited to individual plants but can spread to other nearby plants depending upon climate and soil conditions (i.e., soil moisture) and irrigation practices, resulting in substantial economic losses across large areas (Panth, Hassler & Baysal-Gurel, 2020; Zhang, Yu & Wang, 2021). Phytophthora capsici, various Fusarium spp., Pythium spp., Rhizoctoniasolani, and Macrophomina phaseolina are well-known soilborne pathogens that frequently contribute to yield losses in vegetable cultivation (Panth, Hassler & Baysal-Gurel, 2020; El-kazzaz et al., 2022). Among these common pathogens, Fusarium wilt is a rapidly spreading disease that severely impacts the quality and productivity of pepper plants, similar to its effects on crops such as cotton, cucumber, chickpea, banana (Pu et al., 2022), necessitating effective management.

Cultural measures (such as crop rotation, drip irrigation, and soil fumigation) and chemical methods are utilized to mitigate economic losses caused by soil-borne diseases (Devi et al., 2022). However, chemical methods are now prohibited due to their detrimental effects on the environment and human health. While cultural measures effectively control disease spread, there can be challenges in implementing them. Soil fumigation is limited due to cost, applicability in large areas, and negative impact on beneficial soil microflora. Factors like ozone layer depletion, climate change, drought, and the development of resistance in diseases and pests to chemicals have prompted producers to explore alternative solutions (Li et al., 2019; Ramírez-Gil & Morales-Osorio, 2020; Devi et al., 2022).

The use of resistant varieties or biological control and protective agents is the most secure in the fight against the disease (Yücel & Özarslandan, 2014; Panth, Hassler & Baysal-Gurel, 2020).

In recent times, the utilization of arbuscular mycorrhizal fungus has gained prominence as a biological control strategy in combating bacterial, viral, fungal, and nematode infections across various plant species (Dey & Ghosh, 2022; Weng et al., 2022).

Arbuscular mycorrhizal fungi (AMF) can have a symbiotic relationship with around 71–90% of plants (Song et al., 2015; Dowarah, Gill & Agarwala, 2022; Dey & Ghosh, 2022; Weng et al., 2022). In sustainable agriculture, AMF is recognized as a biofertilizer and biocontrol agent (Dowarah, Gill & Agarwala, 2022). AMF enhances the nutrient uptake of plants (Dowarah, Gill & Agarwala, 2022). Plants also benefit from AMF in their fight against biotic and abiotic stressors (Wu et al., 2021).

AMFs have been tested against both above-ground and soil-borne diseases. The application of Glomus versiforme mycorrhizae to Salvia miltiorrhiza plants was found to enhance the plant’s resistance to the soil-borne pathogen Fusarium oxysporum. The increased protection of mycorrhizal-preinoculated plants against the pathogen was associated with elevated levels of defense enzymes in the roots, such as phenylalanine ammonia-lyase (PAL), chitinase, and β-1,3-glucanase, following pathogen attack (Pu et al., 2022). Song et al. (2015) investigated the effect of Funneliformis mosseae AMF on early blight disease in tomatoes, caused by the pathogen Alternaria solani. They found that AMF application significantly reduced the severity of the disease. Similarly, for this leaf disease, AMF-treated plants exhibited increased levels of defense enzymes—including β-1,3-glucanase, chitinase, PAL, and lipoxygenase (LOX)—in the leaves following pathogen inoculation. This increase in enzyme levels was associated with the successful reduction of the disease. Several studies have also reported how the change in the quantity and quality of plant root secretions following the application of AMF led to a decrease in the number of pathogens in the root zone (Dowarah, Gill & Agarwala, 2022). The quality and quantity of root secretions might vary based on the type of AMF, the plant, and the root colonization (Dowarah, Gill & Agarwala, 2022). More recent reviews provide a more comprehensive summary of studies on the use of several AMF species against various fungal diseases in diverse plants (Dowarah, Gill & Agarwala, 2022; Dey & Ghosh, 2022; Weng et al., 2022).

The interaction between plants, diseases, and AMF in enhancing plant protection through AMF colonization is complex. The effectiveness of various mycorrhizal fungi can differ, and not all AMF will have the same impact on plant defense. Recent research indicates that the benefits of AMF are influenced by factors such as the type of fungal species, the host plant, as well as soil and environmental conditions (Wu et al., 2021; Boutaj et al., 2022). Aside from these, rhizosphere chemistry, environmental circumstances, and interactions with other microorganisms in the mycorrhizal zone are among the factors influencing AMF’s ability to protect plants (Dowarah, Gill & Agarwala, 2022). The application of mycorrhizae alone or in combination with other mycorrhizae or other biological control microorganisms influences their efficiency (Dowarah, Gill & Agarwala, 2022). Other factors include the amount and timing of AMF inoculation, as well as interactions between AMF and host plants (Weng et al., 2022).

Numerous studies have explored the effects of different AMF species on soil-borne diseases caused by various pathogens, investigating the interactions between mycorrhiza, plants, and pathogens through various parameters (Song et al., 2015; Aljawasim, Khaeim & Manshood, 2020; Wu et al., 2021). However, there is limited research on the impact of AMF inoculation on the defense mechanisms of pepper plants against soil-borne pathogens, particularly in relation to Fusarium spp. (Rodriguez-Heredia et al., 2020; Coskun, Alptekin & Demir, 2023). The Güneydoğu Anadolu Projesi (GAP) region, where the study was performed, faces specific challenges due to monoculture and traditional irrigation methods that promote soil-borne pathogen spread (Bilgili, 2017; Bilgili et al., 2018). Despite the known benefits of AMF in enhancing plant resilience and disease control (Bilgili & Güldür, 2018), comprehensive studies addressing their effectiveness in this region remain scarce. Previous research has highlighted the need for detailed investigations into the mechanisms of AMF interactions and signaling processes with plants, but significant gaps and limited applications persist (Pu et al., 2022).

To address these gaps, this study seeks to test the following hypotheses: (1) AMF inoculation significantly enhances the defense mechanisms of pepper plants against soil-borne pathogens, particularly Fusarium spp., compared to non-inoculated plants; (2) The effectiveness of AMF in disease management varies depending on the specific AMF species and their interactions with the soil environment and pepper plants.

The overall goals of this study were to investigate the effects of three different AMFs—Funneliformis mosseae (FM), Rhizophagus intraradices (RI), and Claroideoglomus etunicatum (CE)—as well as a mycorrhizal mix (MM: FM+CE+RI), used as a biological control agent, on two isolates of soil-borne pathogens (Fusarium solani and Fusarium mix) in pepper plants. We examined these effects both independently and in combination to identify the most effective AMF species for disease suppression. Additionally, the study aimed to assess how these AMFs influence plant growth, nutrient uptake, and the rhizosphere of infected plants. By achieving these objectives, the research seeks to enhance our understanding of AMF efficacy in managing soil-borne pathogens and improving pepper plant health.

Materials and Methods

Potting mixture

Imported white sphagnum sterile peat, with an EC value of 35 mS/m (+/− 25%), a pH range of 5.5−6.5, and the fertilizer content amounting to 1.0 kg/m3 with an NPK ratio of 14:10:18, was used as the plant growth medium in pots (TS 1; Klasmann-Deilmann GmbH, Geeste, Germany). Furthermore, before being employed in the study, the peat was re-sterilized in an autoclave at 134 °C for 20 min. To enhance its moisture holding capacity, perlite, which has a high moisture retention ability, was added to the peat in a 1:1 ratio. (Portions of this text were previously published as part of a preprint; https://doi.org/10.21203/rs.3.rs-3260167/v1.)

Plant material & AMFs

The pepper variety INAN-3363 F1 (Capsicum annuum L.), registered by the GAP Agricultural Research Institute (GAPTAEM), was used as the plant material in this study. AMF used in the study; Claroideoglomus etunicatum (syn. Glomus etunicatum; Sensoy et al., 2007), Rhizophagus intraradices (syn. Glomus intraradices; Demir & Onoğur, 1999), and Funneliformis mosseae (syn. Glomus mosseae; Demir et al., 2015), were obtained from Dr. Semra Demir (Van-Yüzüncüyil University, Van, Turkey). The three species were combined in autoclaved sterile peat to create a mycorrhizal mix. The inoculum of the mycorrhizal mix was prepared in a ratio of CE/FM/RI (1:1:1) using sterile peat. The mycorrhizal fungus inoculum was applied at a rate of 1,000 spores/10 g of peat soil (Menge & Timmer, 1982).

Pathogen isolates

The fungal isolates Fusarium solani and Fusarium mix (consisting of F. solani, F. oxysporum, and F. oxysporum f.sp. vasinfectum) were obtained from a survey conducted in the pepper production areas of the GAP region as part of the projects TAGEM-BS-13/09-03/02-06 and HÜBAK-13168 (Bilgili, 2017). These isolates were used in the study after assessing their virulence. Four specific fungal isolates were included in the study: Fusarium mix-50 (Diyarbakır-Cermik), Fusarium mix-147 (Batman-Hasankeyf), Fusarium solani-18 (Sanliurfa-Kisas), and Fusarium solani-48 (Diyarbakir-Yenisehir).

Molecular diagnosis of isolates of fungal pathogens

The morphologically diagnosed fungal isolates were identified at the species level with molecular studies and analyses. DNA isolation was performed using the Plant Genomic DNA Purification Protocol of the Thermo Scientific GeneJET Plant DNA Purification Mini Kit (K0791, K0792; Thermo Fisher Scientific, Waltham, MA, USA). The amplification was performed at 658 bp and 58 °C (Tables 1 and 2). The ITS-Fuf-r primer (Abd-Elsalam et al., 2003) was used for Fusarium spp., and TEF1-α gene primers (Arif et al., 2012; TEF-Fs4f: ATCGGCCACGTCGACTCT and TEF-Fs4r: GGCGTCTGTTGATTGTTAGC) were used specifically for F. solani.

Table 1 Primers used in molecular analysis studies.

Pathogen	Target region	Primer (forward and reverse)	Band length—annealing temperature	Reference	
Fusarium spp.	ITS region	ITS-Fu-f (CAACTCCCAAACCCCTGTGA)	398 bp	Abd-Elsalam et al. (2003)	
ITS-Fu-r (GCGACGATTACCAGTAACGA)	54 °C	
Fusarium oxysporum	Calmodulin gene	CLOX1 (CAGCAAAGCATCAGACCACTATAACTC)	534 bp	Mule et al. (2004)	
CLOX2 (CTTGTCAGTAACTGGACGTTGGTACT)	60 °C	
ITS region	Fov1-Egf (CCACTGTGAGTACTCTCCTCG)	438 bp	Abd-Elsalam et al. (2006)	
Fov1-Egr (CCCAGGCGTACTTGAAGGAAC)	53 °C	
Fusarium solani	TEF1-α gene	TEF-Fs4f (ATCGGCCACGTCGACTCT)	658 bp	Arif et al. (2012)	
TEF-Fs4r (GGCGTCTGTTGATTGTTAGC)	58 °C	
ITS region	ITS-Fu2f (CCAGAGGACCCCCTAACTCT)	595 bp	Arif et al. (2012)	
ITS-Fu2r (CTCTCCAGTTGCGAGGTGTT)	63.5 °C	
ITS2/28S rDNA	ITS-Fs5f (CGTCCCCCAAATACAGTGG)	485 bp	Arif et al. (2012)	
ITS-Fs5r (TCCTCCGCTTATTGATATGCT)	61 °C	

Table 2 PCR results of Fusarium samples used in this study.

DNA sample number	Survey isolate number	Fusarium spp.
Fuf	F. solani	F.oysporum f.sp. vasinfectum
Fov1egr	F. oxysporum
Clox	
Fs5	Fu2	
2	18	+	+	+			
18	147	+		+	+	+	
28	50	+		+	+	+	
26	48	+	+	+			

Experimental design

Pepper seedlings were grown in a medium composed of a 1:1 ratio of peat and perlite, with vermiculite as a covering material, placed in plastic viols (45 cm diameter, 5 cm mesh diameter, 6 cm mesh depth) (Aslanpay & Demir, 2015). Mycorrhizal inoculation was applied by adding 2.5 g of inoculum to the seed bed, while control viols remained uninoculated. Pepper seeds were pre-treated by soaking overnight, washing three times with distilled water, soaking in 2% NaClO for 5 min, and rinsing twice with sterile distilled water before sowing 1 day before the scheduled date. Seedlings were then transplanted into 16 × 18 cm plastic pots with 2–2.5 kg of growing medium. The experiment, conducted in the GAPTAEM Department of Plant Health’s laboratory and climate room, used a randomized plot design with six treatments (FM, CE, RI, MM, Control P, Control N), four replications, and 16 plants per treatment, totaling 96 plants per pathogen. Conditions included 12 h of light per day, 25 °C temperature, and 50–60% relative humidity. Watering was done with distilled water every 2 days until germination and daily thereafter (Vosatka & Gryngler, 1999). After 8 weeks, the mycorrhizal fungi activity was evaluated by measuring root colonization rates and disease severity (Fig. 1).

Figure 1 (A) Root structure of the treatments for 18-F. solani pathogen at the end of the trial (from left to right; Control N; Control P; FM+P; RI+P; CE+P; MM+P). (B) Roots of Control N. (C) Roots of Control P.

Pathogen inoculation into plants

Fungal isolates of Fusarium solani and mixed infections, referred to as Fusarium mix, were collected during a 2014 survey in the pepper production areas of the GAP region. These isolates, which exhibited high virulence, were transferred to PDA nutrient medium in a sterile cabinet. The pathogenic fungal isolates were then placed in 5 mm diameter erlenmayer flasks and incubated for 10 days in a 26 ± 1 °C incubator, following the preparation and autoclaving of an artificial oat culture medium. The spores of the pathogens were adjusted to a density of 1 × 106 CFU/g using the Thoma Lamel before being applied to the plants. Subsequently, the developed medium was inoculated around the roots of the plants by contaminating the potting soil during the transfer of plants from vials to pots. The growth of the pathogens and the formation of lesions were monitored daily in a climate room with 12 h of illumination, at a temperature of 25 ± 1 °C, and a humidity range of 50–60% (Ahmed, Sanchez & Candela, 2000; Alejo-Iturvide et al., 2008; Varma et al., 2009).

Determination of morphological growth parameters of seedlings

The morphological growth parameters of the seedlings were assessed. This included measuring the shoot and root lengths (in cm), the root collar diameter (in mm), the shoot and root fresh weights after harvest (in grams), and the number of leaves per pepper seedling during the harvest. The wet weights of the root and green parts of the plants were measured and then placed in paper bags. These samples were dried in ovens at 70 °C for 48 h to obtain the shoot and root dry weights (Kacar & Inal, 2008).

Determination of micro and macro nutrient contents

The micro and macro nutrient contents were determined separately in both the root zone and shoot of the plants. After harvesting the pepper plants, they were dried and ground. From the ground plant samples, 1 g was taken and subjected to the combustion process (Jones et al., 1991). The filtrates obtained after combustion, using a mixture of nitric and perchloric acid, were used to determine the contents (in mg/kg) of Calcium (Ca), Magnesium (Mg), Phosphorus (P), Potassium (K), Iron (Fe), Manganese (Mn), Zinc (Zn), Sodium (Na), Boron (B), and Copper (Cu) using an ICP-OES device (Jones et al., 1991). For total nitrogen analysis, plant samples underwent three stages: combustion, distillation, and titration, following the procedure described by Kacar & Inal (2008).

Determination of EC and pH in the growing medium

In order to determine the electrical conductivity (EC) and pH values of the growing medium in the different applications, 100 g samples of the potting mixture were weighed and used to prepare saturation paste with distilled water. The pH and EC values of the saturation paste were measured using a pH meter and an EC meter, respectively.

Evaluation of disease severity

Disease symptoms that appeared in plants after the fourth week of pathogen inoculation were evaluated weekly to determine the disease severity. A scale ranging from 0 to 4 was used to assess the disease severity caused by Fusarium solani and Fusarium mix at the end of the 4th, 5th, 6th, and 7th weeks following pathogen inoculation. The scale was defined as follows: 0 indicated no symptoms, 1 represented minor color change in the leaf, slight wilting, or deformation, 2 indicated severe yellowing of the leaf, wilting, and stunting, 3 indicated blackening of the stem, and 4 indicated a completely dry or dead plant. The results of these evaluations were calculated using the formula below to determine the disease severity (%) (Wheeler, 1969);

Diseaseseverityindex=∑(scaleclass×numberofplantsineachscaleclass)×100Totalnumberofplants×Thehighestscaleclass

The disease inhibition rate was calculated to assess the effectiveness of each treatment in suppressing the disease using the formula proposed by De Corato et al. (2020);

Diseaseinhibitionrate(%)=C−TC×100

Determination of AMF root colonization

Fixation and staining

The above-ground parts of the pepper plant were cut, and the root and root collar were carefully removed from the soil. The roots, which were separated from the soil, were thoroughly washed under tap water to clean any soil particles adhering to them. Next, 0.5–1 g pieces of the roots were taken and placed in AFA Fixation liquid, which consists of 90 ml of 70% Alcohol, 5 ml of Formaldehyde, and 5 ml of Acetic acid. The roots were kept in this liquid at +4 °C until the dyeing process. The presence of mycorrhizal fungus and the percentage of colonization were determined by staining the roots in AFA liquid using trypan blue. For the dye solution, a mixture of lactic acid (40 ml), glycerin (80 ml), and distilled water (40 ml) was used, containing 0.4% Trypan blue. This dye solution was adapted from the method of Phillips & Hayman (1970) and modified based on Read, Kouckeki & Hodgsen (1976). The GridLine Intersect Method, as described by Giovannetti & Mosse (1980), was utilized to determine the percentage of colonization of AMF fungi in the dyed roots. Additionally, the number of mycorrhizal spores was determined by staining the roots with Trypan Blue and examining them under a microscope, following the procedure outlined by Read, Kouckeki & Hodgsen (1976) (Fig. 2).

Figure 2 The various appearances of AMF fungi in the roots of pepper seedlings under a light microscope.

Evaluation of the efficacy of AMF Mycorrhizae (Root colonization)

By staining the roots with Trypan Blue, we examined the percentage colonization and the number of mycorrhizal spores in the roots of the pots treated with mycorrhiza under a microscope (Read, Kouckeki & Hodgsen, 1976, Fig. 2). To assess the number of mycorrhizal roots colonized and the efficacy of AMFs, the roots were removed from the pots and washed with tap water after harvest. Subsequently, all roots were placed in capped tubes containing 50% ethyl alcohol, which were kept in the dark in a refrigerator. The roots were then taken out, washed with distilled water, and cut to approximately 5 cm in length from the bottom of the root collar. These sections were wrapped in gauze and sealed with a stapler. The samples were returned to the test tubes, and a 10% KOH solution was added, followed by incubation at 25 °C for 4 days. After the 4th day, the cloth packages were opened, and the roots were washed with distilled water. They were then rewrapped with the cloth packages and submerged in a 1% HCl solution for 3 min. Next, the samples were transferred to a 1% Trypan Blue solution and placed on a magnetic stirrer for 3 h at a mild heat. Finally, the samples were removed from the cloth packages, thoroughly washed with distilled water, and placed on a slide for further examination. The samples were then observed using a light microscope at a magnification of 40x (Koske & Gemma, 1989; Yildirim, 2014).

The percentage of root infection was determined using the calculation method described by Giovannetti & Mosse (1980);

Rootinfection(%)=Numberofinfectedroots(pieces)Totalnumberofroots(pieces)×100

Mycorrhiza total spore count (number/10)

In determining the total number of spores in the soil; samples were taken from the peat perlite mixture in the pots, totaling 10 g. Distilled water (100 ml) was added to the samples, which were then passed through sieves with a mesh size of 53–125 µm. The resulting mixture was centrifuged at 3,000 rpm for 20 min. From the supernatant, 10 ml was extracted and brought up to a total volume of 50 ml by adding 50% sucrose. This mixture was subjected to centrifugation once again at 3,000 rpm for 20 min. Subsequently, 5 ml of the centrifuged samples were placed in a petri dish, and the spore density was examined under a microscope at a magnification of 40x (Gerdeman & Nicolson, 1963; Akay & Karaarslan, 2015).

Statistical analyzes

Statistical analyses were conducted using one-way analysis of variance (ANOVA) to assess differences between groups, followed by Tukey’s Multiple Comparison Test to determine which groups differed significantly. These analyses were carried out using the JMP software.

Results

Plant morphological parameters

All morphological parameters were negatively affected by the application of pathogens to plants in four different pathogen groups. The values of morphological parameters in the control positive groups, where pathogen application was made, were lower than those in the control negative groups across all pathogen groups (Tables 3–6). Depending on the pathogen species, biomass reductions ranged from 10% to 54% in the root parts and from 13% to 56% in the aboveground parts (Table 7). Changes in the fresh and dry weights of shoots and roots were found to be statistically significant in all four groups (p < 0.05), while the significance of other parameters varied depending on the pathogen group and mycorrhiza application.

Table 3 Effects of different AMFs on plants in the 18-F. solani pathogen group.

18-F. solani		F	p	Control N	Control P	CE	CE+P	RI	RI+P	FM	FM+P	MM	MM+P	
Plant	SL	1.5	ns	35.24 ± 1.40a	33.67 ± 3.42a	31.74 ± 1.88a	29.88 ± 0.96a	35.37 ± 2.47a	37.17 ± 3.50a	33.14 ± 1.25a	38.00 ± 6.28a	39.88 ± 1.45a	26.85 ± 2.39a	
Morp	RL	1.6	ns	28.99 ± 1.58a	24.38 ± 3.56a	30.75 ± 1.62a	29.39 ± 4.02a	30.67 ± 2.98a	26.30 ± 4.34a	32.26 ± 0.74a	22.26 ± 1.19a	33.45 ± 3.61a	25.95 ± 2.87a	
Param	RCD	1.9	ns	3.98 ± 0.16a	3.95 ± 0.27a	3.67 ± 0.06a	3.77 ± 0.03a	3.79 ± 0.24a	4.11 ± 0.07a	33.9 ± 0.11a	3.82 ± 0.21a	3.81 ± 0.37a	3.24 ± 0.11a	
	LN	0.9	ns	25.29 ± 1.50a	21.37 ± 2.40a	26.00 ± 3.10a	22.33 ± 0.68a	26.75 ± 4.39a	29.75 ± 0.94a	28.56 ± 1.83a	264.1 ± 3.92a	25.85 ± 1.16a	24.66 ± 3.75a	
	SFW	2.2	ns	47.28 ± 1.93ab	34.79 ± 6.44b	47.60 ± 4.52ab	47.01 ± 8.03ab	42.10 ± 8.07ab	60.97 ± 3.46a	49.50 ± 2.11ab	48.50 ± 4.82ab	35.91 ± 4.34b	33.90 ± 6.48b	
	SDW	2.3	*	4.25 ± 0.28ab	2.90 ± 0.61ab	4.98 ± 0.70a	4.51 ± 0.34ab	4.15 ± 0.82ab	4.77 ± 0.27ab	4.4 ± 50.20ab	4.94 ± 0.45a	3.27 ± 0.37ab	2.55 ± 0.78b	
	RFW	3.7	**	6.26 ± 0.74ab	4.08 ± 1.01b	11.54 ± 0.97a	9.36 ± 1.48ab	7.95 ± 2.41ab	6.46 ± 1.14ab	5.85 ± 0.43ab	7.71 ± 0.26ab	8.34 ± 0.18ab	3.85 ± 0.98b	
	RDW	3.5	**	0.87 ± 0.14ab	0.53 ± 0.13ab	1.15 ± 0.10ab	1.03 ± 0.14ab	1.29 ± 0.18a	0.74 ± 0.21ab	0.75 ± 0.06ab	0.90 ± 0.02ab	0.76 ± 0.05ab	0.37 ± 0.16b	
Plant	B	4.6	**	65.70 ± 3.51a	54.42 ± 2.67ab	63.10 ± 1.04ab	53.38 ± 1.47ab	67.36 ± 2.06a	55.50 ± 2.08ab	62.00 ± 0.85ab	51.01 ± 0.77b	68.064 ± 4.71a	58.16 ± 2.09ab	
Nutr.	Ca	0.7	ns	13,156 ± 1,465a	13,286 ± 478a	13,650 ± 338a	12,447 ± 738a	13,244 ± 424a	15,000 ± 620a	14,586 ± 319a	13,134 ± 573a	13,078 ± 1,828a	15,616 ± 484a	
	Cu	5.2	**	18.55 ± 2.08ab	10.06 ± 1.23b	17.29 ± 0.97ab	8.61 ± 1.04b	22.49 ± 1.75ab	12.82 ± 1.96b	21.02 ± 4.07ab	9.20 ± 0.45b	30.46 ± 6.4a	10.12 ± 0.90b	
	Fe	4.8	**	94 ± 1.95b	128 ± 9.40a	122 ± 3.73a	119 ± 5.35ab	127 ± 11.69a	130 ± 6.37a	137 ± 7.41a	112 ± 2.14ab	120 ± 9.75ab	143 ± 8.51a	
	K	3.8	**	60,396 ± 2,513b	69,560 ± 5,208ab	56,501 ± 2,311b	62,957 ± 4,801ab	65,057 ± 2,153ab	71,184 ± 1,331ab	67,640 ± 2,841ab	58,722 ± 824b	77,046 ± 5,124a	72,183 ± 7,472ab	
	Mg	2.9	**	10,245 ± 982ab	8,172 ± 751ab	9,241 ± 444ab	8,406 ± 1,340ab	8,122 ± 1,408ab	12,520 ± 408a	10,433 ± 315ab	11,362 ± 545ab	6,586 ± 1,379b	10,789 ± 531ab	
	Mn	14.1	**	32 ± 2.84d	47 ± 9.40bcd	57 ± 2.44bc	51 ± 4.87bcd	67 ± 3.55b	91 ± 6.75a	51 ± 1.01bc	42 ± 0.98cd	52 ± 7.41bc	48 ± 0.94bcd	
	Na	9.1	**	361 ± 29.4d	885 ± 60abc	481 ± 57cd	907 ± 83abc	440 ± 68cd	986 ± 40.26ab	566 ± 17.204bcd	847 ± 30.47abc	615 ± 133bcd	1,366 ± 526a	
	P	4.4	**	8,102 ± 416abc	8,068 ± 417abc	7,203 ± 166abc	7,356 ± 538abc	6,425 ± 124c	9,131 ± 532a	8,108 ± 289abc	8,591 ± 322ab	6,771 ± 838bc	9,336 ± 468a	
	Zn	5.9	**	44 ± 4.06b	63 ± 11.7abc	57 ± 5.47abc	60 ± 5.78abc	81 ± 4.83a	71 ± 4.39ac	66 ± 4.53abc	42 ± 1.88bc	83 ± 9.23a	75 ± 5.84abc	
Growth	pH	1.7	ns	6.27 ± 0.02a	6.06 ± 0.14a	6.09 ± 0.01a	6.46 ± 0.09a	5.97 ± 0.29a	6.03 ± 0.03a	6.23 ± 0.09a	6.32 ± 0.25a	6.54 ± 0.15a	6.14 ± 0.12a	
Medium	EC	1.2	ns	0.32 ± 0.03a	0.26 ± 0.01a	0.27 ± 0.07a	0.26 ± 0.03a	0.49 ± 0.14a	0.35 ± 0.02a	0.44 ± 0.15a	0.27 ± 0.01a	0.27 ± 0.01a	0.22 ± 0.01a	
	TN	4.6	**	3.61 ± 0.20abc	3.98 ± 0.32abc	2.85 ± 0.32c	3.89 ± 0.79abc	3.38 ± 0.55bc	5.03 ± 0.24abc	5.96 ± 1.23a	3.69 ± 0.54abc	2.56 ± 0.13c	5.54 ± 0.38ab	
	P	5.9	***	69.60 ± 1.67ab	50.81 ± 0.76bc	34.52 ± 1.32c	62.95 ± 19.7ab	65.17 ± 6.84ab	67.04 ± 1.17ab	64.20 ± 5.14ab	64.93 ± 2.49ab	64.38 ± 3.08ab	76.66 ± 2.96a	
	K	33.9	***	75.70 ± 5.25c	89.00 ± 17.5c	46.20 ± 7.31c	81.50 ± 8.64c	73.70 ± 1.63c	84.30 ± 3.46c	70.87 ± 4.38c	77.60 ± 1.66c	187.30 ± 4.17b	247.70 ± 28.03a	
	Cu	1.9	ns	0.63 ± 0.07a	0.47 ± 0.03a	0.42 ± 0.02a	0.56 ± 0.10a	0.61 ± 0.07a	0.56 ± 0.06a	0.44 ± 0.03a	0.43 ± 0.01a	0.63 ± 0.05a	0.64 ± 0.02a	
	Fe	0.6	ns	14.19 ± 0.43a	14.64 ± 1.24a	16.36 ± 0.14a	15.71 ± 4.72a	18.99 ± 2.52a	16.73 ± 3.42a	14.83 ± 0.98a	11.76 ± 1.0a	16.45 ± 5.24a	17.30 ± 3.37a	
	Mn	2.1	ns	1.61 ± 0.12ab	1.62 ± 0.11ab	1.60 ± 0.08ab	2.05 ± 0.43ab	2.02 ± 0.34ab	1.94 ± 0.41ab	1.38 ± 0.26ab	1.04 ± 0.11b	2.19 ± 0.67ab	2.71 ± 0.36a	
	Zn	0.9	ns	1.19 ± 0.06a	1.26 ± 0.09a	1.34 ± 0.07a	1.61 ± 0.23a	1.68 ± 0.08a	1.61 ± 0.14a	2.32 ± 0.94a	1.14 ± 0.06a	1.79 ± 0.33a	1.83 ± 0.18a	
Note:

SL, Shoot Length (cm); RL, Root Length (cm); RCD, Root CCC Diameter (mm); LN, Number of Leaves; SFW, Shoot Fresh Weight (g); SDW, Shoot Dry Weight (g); RFW, Root Fresh Weight (g); RDW, Root Dry Weight (g). *Significant at p < 0.05; ** Significant at p < 0.01; ns: not significant. Treatment groups with different letters (a, b, c, d or their combinations) are statistically different from each other.

Table 4 Effects of different AMFs on plants in the 48–F. solani pathogen group.

48-F.
solani		F	p	Control N	Control P	CE	CE+P	RI	RI+P	FM	FM+P	MM	MM+P	
Plant	SL	3.8	**	35.24 ± 1.40a	30.24 ± 2.6ab	31.74 ± 1.88ab	27.50 ± 4.35ab	35.37 ± 2.47a	31.30 ± 2.13ab	33.14 ± 1.25a	28.86 ± 2.13ab	39.88 ± 1.45a	20.90 ± 2.85b	
Morp	RL	4.3	**	28.99 ± 1.58a	24.40 ± 4.1ab	30.75 ± 1.62a	24.01 ± 0.83ab	30.67 ± 2.98a	25.54 ± 1.64ab	32.26 ± 0.74a	26.01 ± 4.56ab	33.45 ± 3.61a	13.43 ± 2.34b	
Param	RCD	1.8	ns	3.98 ± 0.16a	3.24 ± 0.34a	3.67 ± 0.07a	3.52 ± 0.26a	3.79 ± 2.24a	3.61 ± 0.14a	3.39 ± 0.11a	3.55 ± 0.26a	3.81 ± 0.37a	3.07 ± 0.07a	
	LN	1.1	ns	25.29 ± 1.50a	16.88 ± 4.72	26.06 ± 3.10a	21.66 ± 3.30a	26.75 ± 4.39a	21.41 ± 3.96a	28.56 ± 1.83a	25.05 ± 6.15a	25.83 ± 1.16a	19.38 ± 2.48a	
	SW	2.2	*	47.28 ± 1.93a	23.92 ± 12.1ab	47.60 ± 4.52a	33.34 ± 8.55ab	42.10 ± 8.07ab	33.76 ± 8.57ab	49.50 ± 2.11a	31.96 ± 12.62ab	35.91 ± 4.34ab	17.65 ± 5.44b	
	SDW	3.1	*	4.25 ± 0.28ab	1.87 ± 0.87b	4.98 ± 0.7a	2.95 ± 0.74ab	4.15 ± 0.82ab	2.93 ± 0.77ab	4.45 ± 0.20ab	2.87 ± 1.12ab	3.27 ± 0.37ab	1.37 ± 0.40b	
	RFW	4.7	**	6.26 ± 0.74ab	4.06 ± 1.3b	11.54 ± 0.97a	5.95 ± 0.46ab	7.95 ± 2.41ab	2.93 ± 0.98b	5.85 ± 0.43ab	4.41 ± 2.18b	8.34 ± 0.18ab	1.59 ± 0.37b	
	RDW	7.2	**	0.87 ± 0.14abc	0.40 ± 0.1cd	1.15 ± 0.10ab	0.53 ± 0.07bcd	1.29 ± 0.18a	0.45 ± 0.09cd	0.75 ± 0.06abcd	0.44 ± 0.21cd	0.76 ± 0.05abcd	0.12 ± 0.03d	
Plant	B	1.9	ns	65 ± 3.51a	50 ± 1.9a	63 ± 1.04a	59 ± 2.7a	67 ± 2.06a	53 ± 1.83a	62 ± 0.85a	57 ± 0.61a	68 ± 4.71a	55 ± 0.91a	
Nutr.	Ca	0.4	ns	13,156 ± 1,465a	15,110 ± 7a	13,650 ± 338a	14,167 ± 2,046a	13,244 ± 424a	14,370 ± 30a	14,586 ± 319a	16,350 ± 1,513a	13,078 ± 1,828a	14,615 ± 1,195a	
	Cu	2.1	*	18.55 ± 2.0ab	14.7 ± 0.4ab	17.29 ± 0.97ab	15.45 ± 4.92ab	22.49 ± 1.75ab	12.82 ± 0.13ab	21.02 ± 4.07ab	10.08 ± 1.08b	30.46 ± 6.40a	12.08 ± 2.29ab	
	Fe	3.8	**	94 ± 1.95b	131 ± 4.2ab	122 ± 3.73a	130 ± 8.10a	127 ± 11.69a	124 ± 5.75ab	137 ± 7.41a	125 ± 5.91ab	120 ± 9.75ab	115 ± 3.35ab	
	K	4.5	**	60,396 ± 25.13bc	68,520 ± 2,340abc	56,507 ± 2,311c	73,942 ± 5,352ab	65,057 ± 2,153abc	72,875 ± 9,675abc	67,640 ± 2,841abc	77,030 ± 5,903ab	77,046 ± 5,124a	82,155 ± 1,175a	
	Mg	1.6	ns	10,245 ± 982a	10,110 ± 89a	9,241 ± 444a	7,371 ± 1,330a	8,122 ± 1,408a	8,734 ± 973a	10,433 ± 315a	11,299 ± 763a	6,586 ± 1,379a	10,188 ± 832a	
	Mn	12.5	**	32 ± 2.84e	74 ± 2.3abcde	57 ± 2.44bcd	79 ± 6.57abc	67 ± 3.55bcd	106 ± 12.64a	51 ± 1.01de	86 ± 22.07ab	52 ± 7.41cde	83 ± 11.76abcd	
	Na	10.9	**	361 ± 29.4d	985 ± 16abcd	481 ± 57cd	1,166 ± 271a	440 ± 68.12d	1,007 ± 13.15abc	566 ± 17.20cd	1,061 ± 66.61ab	615 ± 133bcd	1,297 ± 91a	
	P	2.4	ns	8,102 ± 416a	7,596 ± 62a	7,203 ± 166a	7,530 ± 711a	6,425 ± 124a	7,453 ± 107a	8,108 ± 289a	8,452 ± 548a	6,771 ± 838a	8,925 ± 149a	
	Zn	5.2	**	44.58 ± 4.06b	80.7 ± 1.7ab	57.20 ± 5.47ab	71.32 ± 7.69ab	81.73 ± 4.83a	72.8 ± 4.21ab	66.4 ± 4.53ab	59.5 ± 8.11ab	83 ± 9.23a	78.5 ± 9.64ab	
Growth	pH	1.7	ns	6.26 ± 0.02a	6.41 ± 0.15a	6.08 ± 0.01a	5.91 ± 0.06a	5.97 ± 0.51a	6.13 ± 0.39a	6.22 ± 0.09a	6.58 ± 0.14a	6.54 ± 0.15a	6.51 ± 0.24a	
Med	EC	0.9	ns	0.31 ± 0.03a	0.34 ± 0.01a	0.27 ± 0.07a	0.26 ± 0.02a	0.49 ± 0.14a	0.26 ± 0.02a	0.44 ± 0.15a	0.28 ± 0.003a	0.26 ± 0.02a	0.45 ± 0.18a	
	TN	2.2	ns	3.61 ± 0.20a	4.00 ± a	2.85 ± 0.17a	4.41 ± 1.58a	3.38 ± 0.55a	3.82 ± 0.86a	5.96 ± 1.23a	4.16 ± 1.0a	2.56 ± 0.13a	5.62 ± 0a	
	P	2.7	*	69.59 ± 1.67a	63.44 ± 14.8ab	34.52 ± 1.32b	63.88 ± 5.11ab	65.17 ± 6.84ab	62.86 ± 5.84ab	64.19 ± 5.14ab	63.85 ± 9.59ab	64.37 ± 3.08ab	80.22 ± 9.97a	
	K	11.9	**	75.70 ± 5.25cd	138.70 ± 26.6ab	46.20 ± 7.31d	83.70 ± 1.58bcd	73.70 ± 1.63cd	118.90 ± 13.4bc	70.87 ± 4.38cd	115.70 ± 19.68bc	187.30 ± 4.17a	106.90 ± 15.8bc	
	Cu	1.9	ns	0.62 ± 0.07a	0.63 ± 0.04a	0.42 ± 0.02a	0.60 ± 0.04a	06.0 ± 0.08a	0.62 ± 0.02a	0.43 ± 0.03a	0.51 ± 0.07a	0.63 ± 0.05a	0.45 ± 0.07a	
	Fe	0.8	ns	14.19 ± 0.43a	16.56 ± 4.2a	16.36 ± 0.14a	206.3 ± 1.92a	18.99 ± 2.52a	15.22 ± 3.65a	14.82 ± 0.98a	15.02 ± 4.22a	16.44 ± 5.24a	12.00 ± 1.82a	
	Mn	1.7	ns	1.61 ± 0.12a	2.66 ± 0.3a	1.60 ± 0.07a	2.36 ± 0.21a	2.02 ± 0.34a	2.09 ± 0.41a	1.38 ± 0.27a	1.59 ± 0.65a	2.19 ± 0.67a	1.33 ± 0.10a	
	Zn	0.8	ns	1.18 ± 0.06a	1.55 ± 0.2a	1.34 ± 0.07a	1.81 ± 0.14a	1.67 ± 0.08a	1.64 ± 0.08a	2.31 ± 0.94a	1.40 ± 0.29a	1.70 ± 0.33a	1.43 ± 0.11a	
Note:

SL, Shoot Length (cm); RL, Root Length (cm); RCD, Root CCC Diameter (mm); LN, Number of Leaves; SFW, Shoot Fresh Weight (g); SDW, Shoot Dry Weight (g); RFW, Root Fresh Weight (g); RDW, Root Dry Weight (g). *Significant at p < 0.05; ** Significant at p < 0.01; ns: not significant. Treatment groups with different letters (a, b, c, d or their combinations) are statistically different from each other.

Table 5 Effects of different AMFs on plants in the 50-F.mix pathogen group.

50-F.
mix		F	p	Control N	Control P	CE	CE+P	RI	RI+P	FM	FM+P	MM	MM+P	
Plant	SL	4.1	**	35.24 ± 1.40a	32.21 ± 3.19a	31 ± 1.88ab	28.4 ± 1.45ab	35.37 ± 2.47a	32.4 ± 0.77ab	33.14 ± 1.25a	32.14 ± 2.25ab	39.8 ± 1.45a	21.66 ± 2.18b	
Morp	RL	1.6	ns	28.99 ± 1.58a	24.28 ± 4.30a	30.7 ± 3.25a	30.03 ± 3.0a	30.67 ± 2.98a	30.30 ± 3.52a	32.26 ± 0.74a	24.81 ± 1.90a	33.4 ± 3.61a	23.60 ± 1.60a	
Param	RCD	2.3	*	3.98 ± 0.16a	3.39 ± 0.32ab	3.6 ± 0.07ab	3.79 ± 0.16ab	3.79 ± 0.24ab	3.69 ± 0.10ab	3.39 ± 0.11ab	3.78 ± 0.38ab	3.8 ± 0.37ab	2.82 ± 0.12b	
	LN	1.0	ns	25.29 ± 1.50a	23.0 ± 3.62a	26.0 ± 3.10a	23.5 ± 1.88a	26.75 ± 4.39a	30.0 ± 2.98a	28.56 ± 1.83a	24.0 ± 3.46a	25.8 ± 1.16a	18.33 ± 3.92a	
	SFW	4.2	**	47.28 ± 1.93a	33.99 ± 7.97ab	47.6 ± 4.52a	38.8 ± 7.21ab	42.10 ± 8.07a	54.49 ± 8.60a	49.50 ± 2.11a	42.24 ± 7.61a	35 ± 4.34ab	10.21 ± 4.72b	
	SDW	3.4	**	4.25 ± 0.28a	3.11 ± 0.72ab	4.98 ± 0.70a	3.53 ± 0.48ab	4.15 ± 0.82a	4.53 ± 0.92a	4.45 ± 0.20a	3.80 ± 0.77ab	3.2 ± 0.37ab	0.80 ± 0.31b	
	RFW	4.3	**	6.26 ± 0.74ab	5.46 ± 1.32b	11.5 ± 0.97a	7.80 ± 1.03ab	7.95 ± 2.41ab	3.30 ± 1.47b	5.85 ± 0.43ab	7.13 ± 1.51ab	8.3 ± 0.18ab	1.70 ± 0.32b	
	RDW	4.2	**	0.87 ± 0.14a	0.78 ± 1.15ab	1.15 ± 0.10a	0.71 ± 0.16ab	1.29 ± 0.18a	0.67 ± 0.16ab	0.75 ± 0.06ab	0.68 ± 0.16ab	0.7 ± 0.05ab	0.15 ± 0.03b	
Plant	B	0.7	ns	65.7 ± 3.51a	66.5 ± 3.71a	63.2 ± 0.90a	63.6 ± 2.36a	67.3 ± 2.06a	61.8 ± 1.80a	62.0 ± 0.85a	65.1 ± 2.92a	68.0 ± 4.71a	68.4 ± 1.8a	
Nutr.	Ca	2.8	**	13,156 ± 1,465b	12,291 ± 1,062b	13,707 ± 301b	14,190 ± 858ab	13,244 ± 424a	15,533 ± 238ab	14,586 ± 319ab	13,670 ± 739b	13,078 ± 1828b	20,310 ± 179a	
	Cu	1.5	ns	18.5 ± 2.08a	16.3 ± 1.01a	17.1 ± 0.68a	28.8 ± 10.43a	22.4 ± 1.75a	24.8 ± 2.95a	21.0 ± 4.07a	27.2 ± 7.29a	30.4 ± 6.4a	23.3 ± 0.6a	
	Fe	8.9	**	94 ± 1.95c	101 ± 2.84bc	124 ± 3.0ab	129 ± 4.42ab	127 ± 11.69ab	122 ± 0.77ab	137 ± 7.41a	99 ± 2.70bc	120 ± 9.75abc	153 ± 2.1a	
	K	12.6	**	60,396 ± 2,513cd	59,466 ± 1,987cd	58,177 ± 1,871d	70,678 ± 1,891bc	65,057 ± 2,153bcd	74,811 ± 1,702b	67,640 ± 2,841bcd	68,216 ± 2,895bcd	77,046 ± 5,124b	96,950 ± 1,291a	
	Mg	1.4	ns	10,245 ± 982a	8,017 ± 1,173a	9,314 ± 404a	28,074 ± 1,976a	8,122 ± 1,408a	11,170 ± 225a	10,433 ± 315a	9,465 ± 572a	6,586 ± 1,379a	12,070 ± 145a	
	Mn	22.7	**	32.3 ± 2.84e	55.0 ± 4.22cd	57.6 ± 1.73cd	70.3 ± 6.49bc	67.6 ± 3.55bc	92.0 ± 4.57a	51.2 ± 1.01d	54.5 ± 2.30cd	52.5 ± 7.41cd	81.9 ± 1.2ab	
	Na	11.4	**	361 ± 29.42c	310 ± 18.83c	528 ± 44.78bc	497 ± 49.65bc	440 ± 68.12bc	455b ± 22.07c	566 ± 17.2b	562 ± 45.41bc	615 ± 133b	1147 ± 28a	
	P	1.4	ns	8,102 ± 416a	6,698 ± 556a	7,310 ± 137a	7,822 ± 220a	6,425 ± 124a	7,702 ± 200a	8,108 ± 289a	19,505 ± 1,266a	6,771 ± 838a	10,100 ± 88a	
	Zn	7.8	**	44.5 ± 4.06d	54.0 ± 4.21bcd	55.5 ± 3.78cd	76.3 ± 12.03abc	81.7 ± 4.83a	87.1 ± 7.57a	66.4 ± 4.53abcd	58.9 ± 4.31abcd	83.7 ± 9.23ab	90.3 ± 1.7a	
Grow	pH	3.4	**	6.26 ± 0.02ab	6.13 ± 0.24ab	6.08 ± 0.01ab	6.04 ± 0.01ab	5.97 ± 0.29ab	5.61 ± 0.03b	6.22 ± 0.09ab	6.27 ± 0.12ab	6.54 ± 0.15a	6.61 ± 0.29a	
Med	EC	2.2	ns	0.31 ± 0.03a	0.39 ± 0.10a	0.27 ± 0.07a	0.28 ± 0.03a	0.49 ± 0.14a	0.70 ± 0.09a	0.44 ± 0.15a	0.44 ± 0.05a	0.26 ± 0.02a	0.29 ± 0.06a	
	TN	4.2	**	3.61 ± 0.20ab	2.92 ± 0.32b	2.85 ± 0.32b	3.99 ± 0.53ab	3.38 ± 0.55ab	5.01 ± 0.60ab	5.96 ± 1.23a	4.93 ± 0.93ab	2.56 ± 0.13b	5.48 ± ab	
	P	4.9	**	69.59 ± 1.67a	63.77 ± 11.99a	34.52 ± 1.32b	49.07 ± 6.09ab	65.17 ± 6.84a	74.23 ± 1.20a	64.19 ± 5.14a	61.79 ± 6.21a	64.37 ± 3.08a	71.30 ± 2.95a	
	K	18.3	**	73.70 ± 5.25cde	108.80 ± 17.90cd	46.20 ± 7.31e	54.00 ± 9.08de	73.70 ± 1.63cde	124.60 ± 16.4bc	70.87 ± 4.38cde	75.80 ± 1.31cde	187.30 ± 4.17a	176.90 ± 27.03ab	
	Cu	2.7	ns	0.62 ± 0.07a	0.55 ± 0.04a	0.42 ± 0.02a	0.38 ± 0.04a	0.60 ± 0.08a	0.63 ± 0.03a	0.43 ± 0.03a	0.46 ± 0.03a	0.63 ± 0.05a	0.54 ± 0.04a	
	Fe	1.8	ns	14.19 ± 0.43a	18.33 ± 2.05a	16.36 ± 0.14a	15.40 ± 0.89a	18.99 ± 2.52a	21.52 ± 1.12a	14.82 ± 0.98a	13.42 ± 2.14a	16.44 ± 5.24a	18.96 ± 1.48a	
	Mn	3.4	**	1.61 ± 0.12b	2.55 ± 0.35ab	1.60 ± 0.07ab	1.48 ± 0.12ab	2.02 ± 0.35ab	2.96 ± 0.11a	1.38 ± 0.26b	1.25 ± 0.08b	2.19 ± 0.67ab	2.00 ± 0.39ab	
	Zn	0.9	ns	1.18 ± 0.06a	1.53 ± 0.08a	1.34 ± 0.06a	1.38 ± 0.18a	1.67 ± 0.08a	1.97 ± 0.02a	2.31 ± 0.94a	1.29 ± 0.11a	1.70 ± 0.33a	1.66 ± 0.14a	
Note:

SL, Shoot Length (cm); RL, Root Length (cm); RCD, Root CCC Diameter (mm); LN, Number of Leaves; SFW, Shoot Fresh Weight (g); SDW, Shoot Dry Weight (g); RFW, Root Fresh Weight (g); RDW, Root Dry Weight (g). *Significant at p < 0.05; ** Significant at p < 0.01; ns: not significant. Treatment groups with different letters (a, b, c, d, e or their combinations) are statistically different from each other.

Table 6 Effects of different AMFs on plants in the 147–F.mix pathogen group.

147-F
.mix		F	p	Control N	Control P	CE	CE+P	RI	RI+P	FM	FM+P	MM	MM+P	
Plant	SL	1.7	ns	35.24 ± 1.40a	30.77 ± 2.01a	31.74 ± 1.88a	26.27 ± 1.72a	35.37 ± 2.47a	27.69 ± 1.35a	33.14 ± 1.25a	31.94 ± 9.96a	39.88 ± 2.05a	23.53 ± 1.88a	
Mor	RL	2.2	ns	28.99 ± 1.58a	24.81 ± 2.61a	30.75 ± 1.62a	29.20 ± 2.51a	30.67 ± 2.98a	22.60 ± 2.84a	32.26 ± 0.74a	21.55 ± 7.75a	33.45 ± 3.61a	20.70 ± 1.97a	
Par.	RCD	3.9	**	3.98 ± 0.16a	3.88 ± 0.22ab	3.67 ± 0.07ab	3.11 ± 0.25ab	3.79 ± 0.25ab	3.33 ± 0.03ab	3.39 ± 0.12ab	4.36 ± 0.81a	3.81 ± 0.37ab	2.65 ± 0.24a	
	LN	1.1	ns	25.29 ± 1.50a	24.83 ± 3.75a	26.00 ± 3.10a	14.00 ± 0.80a	26.75 ± 4.39a	16.91 ± 1.08a	28.56 ± 1.83a	30.16 ± 13.74a	25.83 ± 1.16a	19.77 ± 5.30a	
	SFW	2.9	*	47.28 ± 1.93ab	40.94 ± 7.19ab	47.60 ± 4.52ab	18.72 ± 1.11b	42.10 ± 8.07ab	24.17 ± 3.93ab	49.50 ± 2.11a	36.23 ± 17.57ab	35.91 ± 4.34ab	19.73 ± 8.51b	
	SDW	2.9	*	4.25 ± 0.28ab	3.21 ± 0.56ab	4.98 ± 0.70a	2.21 ± 0.42ab	4.15 ± 0.82ab	2.00 ± 0.23b	4.45 ± 0.20ab	3.26 ± 1.55ab	3.27 ± 0.37ab	1.56 ± 0.69b	
	RFW	4.3	**	6.26 ± 0.74ab	4.33 ± 1.03b	11.54 ± 0.97a	8.11 ± 0.09ab	7.95 ± 2.41ab	2.11 ± 0.77b	5.85 ± 0.43ab	3.82 ± 1.91b	8.34 ± 0.18ab	4.43 ± 1.76ab	
	RDW	5.4	**	0.87 ± 0.14abc	0.61 ± 0.13bc	1.15 ± 0.10ab	0.69 ± 0.07abc	1.29 ± 0.18a	0.30 ± 0.08c	0.75 ± 0.06abc	0.77 ± 0.07abc	0.76 ± 0.06abc	0.35 ± 0.12c	
Plant	B	0.9	ns	65.7 ± 11.64a	65.2 ± 3.67a	63.1 ± 1.04a	67.2 ± 4.78a	67.3 ± 2.06a	73.8 ± 5.19a	62.0 ± 0.85a	64.7 ± 1.06a	68.0 ± 4.71a	65.4 ± 0.93a	
Nutr.	Ca	1.3	ns	13,156 ± 1,465a	15,611 ± 597a	13,650 ± 338a	15,960 ± 1,748a	13,244 ± 424a	13,812 ± 418a	14,586 ± 319a	15,875 ± 523a	13,078 ± 1,828a	16,876 ± 433a	
	Cu	2.6	*	18.5 ± 2.08ab	16.97 ± 1.23ab	17.29 ± 0.97ab	15.66 ± 1.21ab	22.49 ± 1.75ab	14.7 ± 0.68ab	21.02 ± 4.07ab	10.80 ± 2.26b	30.46 ± 6.40a	21.19 ± 2.55ab	
	Fe	4.4	**	94 ± 1.95b	112 ± 6.37ab	122 ± 3.73a	117 ± 10.0ab	127 ± 11.69a	131 ± 9.84a	137 ± 7.41a	102 ± 1.59ab	120 ± 9.75ab	130 ± 7.92ab	
	K	6	**	60,396 ± 2,513bcd	74,268 ± 3,812abc	56,501 ± 2,311cd	78,400 ± 6,270abc	65,057 ± 2,153bcd	80,785 ± 4,681ab	67,640 ± 2,841abcd	46,149 ± 15,656d	77,046 ± 5,124abc	92,920 ± 8,262a	
	Mg	3.5	**	10,245 ± 982ab	10,197 ± 929ab	9,241 ± 444ab	5,397 ± 615b	8,122 ± 1,408ab	7,599 ± 952ab	10,433 ± 315ab	12,610 ± 585a	6,586 ± 1,379b	11,333 ± 134ab	
	Mn	21	**	32.3 ± 2.85d	80.2 ± 7.09b	57.5 ± 2.44c	117 ± 14.23a	67.6 ± 3.55bc	89.1 ± 6.72ab	51.2 ± 1.01c	61.8 ± 4.37bc	52.5 ± 7.41cd	84.5 ± 2.56b	
	Na	4.6	**	361 ± 29.42b	674 ± 58.26a	481 ± 57.9ab	705 ± 82.20a	440 ± 68.12ab	710 ± 75.00a	566 ± 17.20ab	716 ± 39.06a	615 ± 133ab	724 ± 132a	
	P	3.0	**	8,102 ± 416a	7,526 ± 554a	7,203 ± 166a	6,625 ± 1,042a	6,425 ± 124a	6,496 ± 121a	8,108 ± 289a	8,578 ± 286a	6,771 ± 838a	8,586 ± 195a	
	Zn	7.2	**	44.5 ± 4.07bc	61.5 ± 7.94abc	57.2 ± 5.47abc	68.3 ± 10.25abc	81.7 ± 4.83a	78.5 ± 8.45a	66.4 ± 4.53ab	31.2 ± 0.80c	83.7 ± 9.23a	83.9 ± 6.07a	
Grow	pH	1.4	ns	6.26 ± 0.02a	5.78 ± 0.19a	60.8 ± 0.01a	6.26 ± 0.12a	5.97 ± 0.29a	6.06 ± 0.41a	6.22 ± 0.09a	6.23 ± 0.13a	6.54 ± 0.15a	6.08 ± 0.07a	
Med	EC	1.3	ns	0.31 ± 0.03a	0.26 ± 0.02a	0.27 ± 0.07a	0.37 ± 0.07a	0.49 ± 0.15a	0.61 ± 0.13a	0.44 ± 0.16a	0.29 ± 0.10a	0.26 ± 0.02a	0.37 ± 0.12a	
	TN	4.5	**	3.61 ± 0.20b	4.46 ± 0.45ab	2.85 ± 0.33b	2.14 ± 0.30b	3.38 ± 0.55b	4.57 ± 0.45ab	5.96 ± 1.23a	3.73 ± 0.46ab	2.56 ± 0.13b	4.18 ± ab	
	P	7.6	**	69.59 ± 1.67a	71.08 ± 9.32a	34.52 ± 1.32b	37.70 ± 3.80b	65.17 ± 6.84a	66.17 ± 5.63a	64.19 ± 5.14a	77.71 ± 7.36a	64.37 ± 3.08a	69.62 ± 4.33a	
	K	3.4	**	75.70 ± 5.25b	114.80 ± 4.75ab	46.20 ± 7.31b	94.20 ± 18.01ab	73.70 ± 1.63ab	134.70 ± 28.10ab	70.87 ± 4.38b	165.10 ± 75.00ab	187.30 ± 4.17a	101.5 ± 20.3ab	
	Cu	2.2	ns	0.62 ± 0.07a	0.68 ± 0.01a	0.42 ± 0.01a	0.59 ± 0.08a	0.60 ± 0.08a	0.67 ± 0.03a	0.43 ± 0.03a	0.44 ± 0.02a	0.63 ± 0.05a	0.36 ± 0.18a	
	Fe	1.3	ns	14.19 ± 0.43a	21.04 ± 4.71a	16.36 ± 0.14a	12.17 ± 2.20a	18.99 ± 2.52a	21.25 ± 3.31a	14.82 ± 0.98a	13.01 ± 0.51a	16.44 ± 5.24a	12.10 ± 6.09a	
	Mn	1.9	ns	1.61 ± 0.12a	2.85 ± 0.47a	1.60 ± 0.08a	1.69 ± 0.13a	2.02 ± 0.35a	2.88 ± 0.72a	1.38 ± 0.26a	1.43 ± 0.20a	2.19 ± 0.67a	1.33 ± 0.72a	
	Zn	0.9	ns	1.18 ± 0.06a	1.66 ± 0.12a	1.34 ± 0.07a	1.60 ± 0.09a	1.67 ± 0.08a	2.00 ± 0.15a	2.31 ± 0.94a	1.25 ± 0.08a	1.70 ± 0.33a	1.69 ± 0.12a	
Note:

SL, Shoot Length (cm); RL, Root Length (cm); RCD, Root CCC Diameter (mm); LN, Number of Leaves; SFW, Shoot Fresh Weight (g); SDW, Shoot Dry Weight (g); RFW, Root Fresh Weight (g); RDW, Root Dry Weight (g). *Significant at p < 0.05; ** Significant at p < 0.01; ns: not significant. Treatment groups with different letters (a, b, c, d or their combinations) are statistically different from each other.

Table 7 Percent damage caused by pathogen inoculation compared to non-inoculated control plants.

		18 F.solani	48 F.solani	50 F.mix	147 F.mix	
Plant Morp	SL	−4.46	−14.19	−8.60	−12.68	
Parameters	RL	−15.90	−15.83	−16.25	−14.42	
	RRM	−0.75	−18.59	−14.82	−2.51	
	LN	−15.90	−33.25	−9.05	−1.82	
	SFW	−26.42	−49.41	−28.11	−13.41	
	SDW	−31.76	−56.00	−26.82	−24.47	
	RFW	−34.82	−35.14	−12.78	−30.83	
	RDW	−39.08	−54.02	−10.34	−29.89	
Plant Nutr	B	−17.17	−23.08	1.22	−0.76	
	Ca	0.99	14.85	−6.57	18.66	
	Cu	−45.77	−20.32	−11.89	−8.52	
	Fe	36.17	39.36	7.45	19.15	
	K	15.17	13.45	−1.54	22.97	
	Mg	−20.23	−1.32	−21.75	−0.47	
	Mn	46.88	131.25	70.28	148.30	
	N	145.15	172.85	−14.13	86.70	
	P	−0.42	−6.25	−17.33	−7.11	
	Zn	43.18	81.02	21.35	38.20	
Growth med	pH	−3.35	2.40	−2.08	−7.67	
	EC	−18.75	9.68	25.81	−16.13	
	TN	10.25	10.80	−19.11	23.55	
	P	−27.00	−8.84	−8.36	2.14	
	K	17.57	83.22	47.63	51.65	
	Cu	−25.40	1.61	−11.29	9.68	
	Fe	3.17	16.70	29.18	48.27	
	Mn	0.62	65.22	58.39	77.02	
	Zn	5.88	31.36	29.66	40.68	
Note:

SL, Shoot Length (cm); RL, Root Length (cm); RCD, Root CCC Diameter (mm); LN, Number of Leaves; SFW, Shoot Fresh Weight (g); SDW, Shoot Dry Weight (g); RFW, Root Fresh Weight (g); RDW, Root Dry Weight (g).

The negative effects of pathogen application on plant growth parameters were mostly counteracted by mycorrhizal application in the 18-F. solani pathogen group and partially in the 50-F.mix and 147-F.mix pathogen groups. Except for a few instances, the application of CE, RI, and FM in the 18-F. solani group was largely effective in reversing the pathogen’s effects on all morphological parameters, whereas the MM was largely ineffective in countering these effects (Table 8). The CE+P group had 49% more root dry weight compared to the pathogen-free control group (Table 8). In the 48-F. solani group, only the FM application (FM+P) was able to eliminate the negative effect of pathogen application on the number of leaves. In the 50-F. mix pathogen group, The CE application (CE+P) effectively countered the negative effects of the pathogen on root length and root fresh weight, showing 3.6% and 24% more root length and root fresh weight compared to the pathogen-free control group (Table 8). The RI application (RI+P) countered the pathogen’s negative effects on root length, as well as shoot fresh and dry weight. The FM application (FM+P) successfully mitigated the pathogen-induced reduction in root fresh weight.

Table 8 Percentage improvement in mycorrhiza-inoculated plants contaminated with pathogen compared to non-inoculated control plants.

		18-F.solani	48-F. solani	50-F.mix	147-F.mix	
		CE+P	RI+P	FM+P	MM+P	CE+P	RI+P	FM+P	MM+P	CE+P	RI+P	FM+P	MM+P	CE+P	RI+P	FM+P	MM+P	
Plant morp	SL	−15.2	5.5	7.8	−23.8	−22.0	−11.2	−18.1	−40.7	−19.4	−8.0	−8.8	−38.5	−25.5	−21.4	−9.4	−33.2	
Parameters	RL	1.4	−9.3	−23.2	−10.5	−17.2	−11.9	−10.3	−53.7	3.6	4.5	−14.4	−18.6	0.7	−22.0	−25.7	−28.6	
	RRM	−5.3	3.3	−4.0	−18.6	−11.6	−9.3	−10.8	−22.9	−4.8	−7.3	−5.0	−29.1	−21.9	−16.3	9.5	−33.4	
	LN	−11.7	17.6	4.4	−2.5	−14.4	−15.3	−0.9	−23.4	−7.1	18.6	−5.1	−27.5	−44.6	−33.1	19.3	−21.8	
	SFW	−0.6	29.0	2.6	−28.3	−29.5	−28.6	−32.4	−62.7	−17.8	15.2	−10.7	−78.4	−60.4	−48.9	−23.4	−58.3	
	SDW	6.1	12.2	16.2	−40.0	−30.6	−31.1	−32.5	−67.8	−16.9	6.6	−10.6	−81.2	−48.0	−52.9	−23.3	−63.3	
	RFW	49.5	3.2	23.2	−38.5	−5.0	−53.2	−29.6	−74.6	24.6	−47.3	13.9	−72.8	29.6	−66.3	−39.0	−29.2	
	RDW	18.4	−14.9	3.4	−57.5	−39.1	−48.3	−49.4	−86.2	−18.4	−23.0	−21.8	−82.8	−20.7	−65.5	−11.5	−59.8	
Plant Nutr	B	−18.8	−15.5	−22.4	−11.5	−9.2	−18.5	−12.3	−15.4	−3.2	−5.9	−0.9	4.1	2.3	12.3	−1.5	−0.5	
	Ca	−5.4	14.0	−0.2	18.7	7.7	9.2	24.3	11.1	7.9	18.1	3.9	54.4	21.3	5.0	20.7	28.3	
	Cu	−53.6	−30.9	−50.4	−45.4	−16.7	−30.9	−45.7	−34.9	55.7	34.1	47.0	25.9	−15.6	−20.8	−41.8	14.2	
	Fe	26.6	38.3	19.1	52.1	38.3	31.9	33.0	22.3	37.2	29.8	5.3	62.8	24.5	39.4	8.5	38.3	
	K	4.2	17.9	−2.8	19.5	22.4	20.7	27.5	36.0	17.0	23.9	12.9	60.5	29.8	33.8	−23.6	53.9	
	Mg	−18.0	22.2	10.9	5.3	−28.1	−14.7	10.3	−0.6	174.0	9.0	−7.6	17.8	−47.3	−25.8	23.1	10.6	
	Mn	59.4	184.4	31.3	50.0	146.9	231.3	168.8	159.4	117.6	184.8	68.7	153.6	262.2	175.9	91.3	161.6	
	N	151.2	173.1	134.6	278.4	223.0	178.9	193.9	259.3	37.7	26.0	55.7	217.7	95.3	96.7	98.3	100.6	
	P	−9.2	12.7	6.0	15.2	−7.1	−8.0	4.3	10.2	−3.5	−4.9	140.7	24.7	−18.2	−19.8	5.9	6.0	
	Zn	36.4	61.4	−4.5	70.5	60.0	63.3	33.5	76.1	71.5	95.7	32.4	102.9	53.5	76.4	−29.9	88.5	
Growth med	pH	3.0	−3.8	0.8	−2.1	−5.6	−2.1	5.1	4.0	−3.5	−10.4	0.2	5.6	0.0	−3.2	−0.5	−2.9	
	EC	−18.8	9.4	−15.6	−31.3	−16.1	−16.1	−9.7	45.2	−9.7	125.8	41.9	−6.5	19.4	96.8	−6.5	19.4	
	TN	7.8	39.3	2.2	53.5	22.2	5.8	15.2	55.7	10.5	38.8	36.6	51.8	−40.7	26.6	3.3	15.8	
	P	−9.6	−3.7	−6.7	10.1	−8.2	−9.7	−8.2	15.3	−29.5	6.7	−11.2	2.5	−45.8	−4.9	11.7	0.0	
	K	7.7	11.4	2.5	227.2	10.6	57.1	52.8	41.2	−26.7	69.1	2.8	140.0	24.4	77.9	118.1	34.1	
	Cu	−11.1	−11.1	−31.7	1.6	−3.2	0.0	−17.7	−27.4	−38.7	1.6	−25.8	−12.9	−4.8	8.1	−29.0	−41.9	
	Fe	10.7	17.9	−17.1	21.9	45.4	7.3	5.8	−15.4	8.5	51.7	−5.4	33.6	−14.2	49.8	−8.3	−14.7	
	Mn	27.3	20.5	−35.4	68.3	46.6	29.8	−1.2	−17.4	−8.1	83.9	−22.4	24.2	5.0	78.9	−11.2	−17.4	
	Zn	35.3	35.3	−4.2	53.8	53.4	39.0	18.6	21.2	16.9	66.9	9.3	40.7	35.6	69.5	5.9	43.2	
Note:

SL, Shoot Length (cm); RL, Root Length (cm); RCD, Root CCC Diameter (mm); LN, Number of Leaves; SFW, Shoot Fresh Weight (g); SDW, Shoot Dry Weight (g); RFW, Root Fresh Weight (g); RDW, Root Dry Weight (g).

Plant nutritions

The Tables 3–6 shows changes in plant nutrient contents following pathogen and mycorrhiza treatments in four different pathogen groups. Ca, Fe, K, Mn, Na, and Zn ratios in plants increased in all groups following pathogen application (p < 0.05). Comparing the control positive groups with pathogen applications to the control negative groups, the levels of these nutrients are higher in the control positive groups. In these nutrients, increases of up to 18%, 39%, 22%, 148%, and 171% were observed after pathogen application compared to the pathogen-free negative control group, respectively (Table 7).

After applying the pathogen, decreases in the contents of the plants were seen in the remaining other nutrients, B, Cu, Mg, and P. In 18-F. solani and 147-F. mix pathogen groups, the reduction in Mg and P contents caused by pathogen application was statistically significant (p < 0.05), but in 48-F. solani and 50-F. mix pathogen groups, it was found to be statistically insignificant (p > 0.05). The decrease in Cu was statistically insignificant (p < 0.05) only in the 50-F.mix pathogen group, but significant (p > 0.05) in the other three.

In all pathogen groups, the decreases in these nutrients following pathogen application were effectively balanced, particularly with MM (MM+P) application. Plants treated with MM application and inoculated with 50-F. mix pathogen showed increases of 54%, 62%, 60%, 17%, 153%, 217%, and 102% in these nutrients, respectively, compared to the negative control group (Table 8). The reductions in these three nutrients in the 18-F. solani group were countered with RI (RI+P), FM (FM+P), and MM (MM+P) applications. The decreases in Mg and P elements in the 48-F. solani group were balanced with FM (FM+P) and MM (MM+P) applications. In the 50-F. mix group, the decreases in Cu and Mg were balanced with CE (CE+P), RI (RI+P), FM (FM+P), and MM (MM+P) applications, while the decrease in P was balanced with FM (FM+P) and MM (MM+P) applications. For the 147-F. mix group, the pathogen-induced decreases in Mg and P were balanced with FM (FM+P) and MM (MM+P) applications, while the decrease in Cu was only balanced with MM (MM+P) application.

Growth medium

In terms of growth medium EC and pH values, there were no statistically significant differences between treatments in different pathogen groups (Tables 3–5) (p > 0.05). However, when comparing the N, P, and K contents, significant differences were found between applications in different pathogen groups (p < 0.05). Following pathogen application, the contents of N and K increased by up to 23% and 83%, respectively, while the content of P decreased by up to 27%, depending on the species of the pathogen (Table 7). For Fe, Cu, Zn, and Mn elements, there were no statistically significant differences between treatments in any of the pathogen groups.

Root colonization

Figure 3 presents the percentage of root colonization achieved through four different AMF applications in various pathogen groups. Statistically significant differences (p < 0.05) in root colonization were observed among different AMF applications in all pathogen groups. The rates of root colonization varied across different pathogen and mycorrhiza application groups, ranging from 33% to 65% in plants treated solely with mycorrhiza and from 53% to 93% in plants where mycorrhiza and pathogen were applied together. The highest average root colonization (93%) was obtained from the MM+P application in the 50-F. mix group (Fig. 3).

Figure 3 The statistical comparison of total spore counts of arbuscular mycorrhizal fungi (AMF) among the four pathogens studied.

In all pathogen groups, for the three mycorrhizal species, plants treated with mycorrhiza and pathogen together (CE+P, RI+P, MM+P) exhibited higher root colonization compared to plant groups treated only with mycorrhiza (CE, RI, MM). Conversely, for the FM species, the plant groups treated solely with mycorrhiza showed higher rates of root colonization than the groups treated with both pathogen and mycorrhiza together, except in the 48-F. solani pathogen group. In the 48-F. solani pathogen group, the FM and FM+P groups displayed an equal level of colonization (Fig. 3).

Total spore counts

Figure 4 display the total number of spores obtained from four different AMF applications in various pathogen groups. Significant statistical differences (p < 0.05) in the total spore count were observed among AMF applications in all pathogen groups. The total spore rates varied across pathogen and mycorrhiza application groups, ranging from 3.63 to 12.66 in plants treated solely with mycorrhiza and from 4.66 to 12.66 in plants treated with both mycorrhiza and pathogen. Among all pathogen groups, the MM application resulted in the highest overall spore count (12.66). Additionally, the FM+P application in the 50-F. mix pathogen group also yielded the highest overall spore count, along with MM (Fig. 4).

Figure 4 The statistical comparison of the percentage of mycorrhizal colonization by arbuscular mycorrhizal fungi (AMF) among the four different pathogens used in the study.

Disease severity

Table 9 provides a comparison of disease severity and disease suppression rates for different pathogen and mycorrhiza treatment groups. With the exception of the 147-F. mix group (p < 0.05), there were no statistically significant differences (p > 0.05) in the severity and suppressiveness of the disease between the applications. However, the severity of the disease decreased in plant groups with mycorrhizae at varying rates compared to plants inoculated with pathogens alone. Despite exhibiting the most severe disease, the 147-F. mix group was followed by the 48-F. solani, 50-F. mix, and 18-F. solani groups, in that order. When examining the disease severity data for the 18-F. solani pathogen, it was found that the pathogen’s disease severity was 20.83%. Among the mycorrhizal applications, CE showed the lowest disease severity rate (13.89%), making it the most effective mycorrhizal treatment. It was followed by RI (16.16%) and FM and MM (19.44%). In terms of disease suppression rate, CE exhibited the highest effectiveness (58.32%). When comparing the groups based on disease suppression rate, CE was followed by RI, FM, and MM. When analyzing the percent disease severity of the 48-F. solani pathogen, the Control (+) group exhibited the highest disease severity after 147-F. mix (58.33%). Among the AMF application groups, the disease severity rates for this pathogen were ranked as follows: FM, MM, RI, and CE, respectively. Among the AMF treatments, CE application (48.14%) was the most effective in suppressing this pathogen, followed by RI (40.74%), MM (29.63%), and FM (29.62%).

Table 9 The differences between the treatment groups in terms of disease severity and disease suppression rates.

	18-F. solani	48-F. solani	50-F. mix	147-F. mix	
Treatments	DS (%)	DI (%)	DS (%)	DI (%)	DS (%)	DI (%)	DS (%)	DI (%)	
Control N	0		0				0		
Control P (Pathogen)	20.83a	–	56.25a	–	35.41a	–	58.33ab	–	
FM+Pathogen	19.44a	41.66a	52.78a	29.62a	38.88a	22.23a	77.77a	−16.67b	
RI+Pathogen	16.16a	49.99a	44.44a	40.74a	36.11a	27.78a	52.77ab	20.82ab	
CE+Pathogen	13.89a	58.32a	38.89a	48.14a	41.66a	16.68a	33.33b	50.00a	
MM+Pathogen	19.44a	41.66a	52.77a	29.63a	66.66a	−33.32a	41.66ab	37.50ab	
Note:

FM, Funneliformis mosseae; RI, Rhizophagus intraradices; CE, Claroideoglomus etunicatum; MM, Combination of all three mycorrhiza species; DS, Disease severity; DI, Disease inhibition. Treatment groups with different letters (a, b or their combinations) are statistically different from each other.

When examining disease severity for the 147-F. mix pathogen, CE was ranked first, followed by MM, RI, and FM. Similarly, when considering the suppression rate of this 147-F. mix pathogen by AMF, the ranking matched the disease severity percentages (DS%). However, there was a negative suppression rate observed in the FM application (−16.67%). Similarly, a negative disease inhibition rate (−) was obtained in the MM application (−33.32) for the 50-F. mix pathogen.

When examining the disease severity rates for the 50-F. mix pathogen, it was observed that the AMF application groups, in contrast to the other three pathogens, increased the severity of the disease rather than decreasing it. The disease severity in the Control (+) group for the 50-F. mix pathogen was 35.41%. In comparison, MM exhibited a severity of 66.66%, CE had 41.66%, FM had 38.88%, and RI had 36.11%. Similarly, the disease suppression rates for MM, CE, FM, and RI were −33.32%, 16.68%, 22.23%, and 27.78%, respectively.

Discussion

After pathogen application, all plant growth parameters, across all pathogen groups, were generally negatively impacted. On the other hand, mycorrhiza application counteracted this pathogen-caused negativity in plants in the 18-F. solani pathogen group, but it was unable to do so in the 48-F. solani pathogen group, where the disease severity is greater. Only a few instances of mycorrhiza application were able to completely reverse the negative effects brought on by the pathogen in the 50-F. mix and 147-F. mix pathogen groups, where the disease severity is comparatively higher. This finding demonstrates that mycorrhizae are ineffective after a certain level of disease severity. While CE, FM, and RI were successful in counteracting the pathogen’s negative effects on morphological parameters, MM composed of their mixture had no effect. Various studies have also found that AMFs have a positive effect on plant growth parameters in soil-borne pathogen-treated plants (Vigo, Norman & Hooker, 2000; Ozgonen & Erkilic, 2007; Hafez et al., 2013; Aljawasim, Khaeim & Manshood, 2020; Wu et al., 2021). In their study, Aljawasim, Khaeim & Manshood (2020) found a significant increase in both shoot dry weight and root dry weight in plants treated with mycorrhiza compared to plants not applied. Demir et al. (2023) found that the application of AMF, specifically FM and Gigaspora margarita (Gm), had significant effects on the morphological parameters of strawberry plants infected with various pathogens. They noted that different AMF treatments resulted in varying increases in plant fresh weight, dry weight, and length, depending on the specific pathogen involved.

In contrast to plant growth parameters, some plant nutrients (Ca, Fe, K, Mn, Na and Zn) increased after pathogen application, which can be attributed to increased plant nutrient intakes in response to stress. Pathogen application resulted in decreases in nutrient elements such as Cu, Mg, and Zn, similar to plant growth parameters, but these decreases were mostly balanced by mycorrhiza application. Following pathogen application, MM, FM, RI and CE were respectively the most effective in maintaining the decrease in these plant nutrients in plants. CE application was rarely found to be effective in maintaining plant nutrient content. AMFs are an important tool for increasing plant nutrient absorption and stress tolerance in biotic stress conditions, as well as for biological control of cucumber plants treated with soil-borne pathogen (R. solani) (Aljawasim, Khaeim & Manshood, 2020). In Verticillium dahliae-inoculated pepper plants, all macro and micronutrients, except for N, were found to be lower when compared to uninoculated control plants. The application of AMF led to a slight increase in P, Mg, Cu, Mn, and B levels. However, this increase was not statistically significant (Coskun, Alptekin & Demir, 2023).

With the exception of the FM+P case, the coexistence of both AMF and pathogens exhibited synergistic effects on root colonization. The presence of pathogens had a positive impact on the development of different AMFs in the root zone, as shown in Fig. 3. The presence of pathogens in soils has the potential to influence the colonization of AMF and the development of AMF in the presence of pathogens can vary depending on biotic and abiotic conditions and is influenced by the interactions between AMF, hosts, and pathogens (Spagnoletti et al., 2021; Coskun, Alptekin & Demir, 2023). While previous studies have reported that pathogen infection reduces mycorrhizal colonization in pepper (Coskun, Alptekin & Demir, 2023), the researchers investigating the interaction between AMF (Rhizophagus intraradices) and a soilborne pathogen (Fusarium pseudograminearum) on the root colonization of wheat plants found that the simultaneous inoculation of AMF and the pathogen resulted in higher AMF colonization percentages compared to AMF alone (Spagnoletti et al., 2021).

In general, pathogen severity was lower in plants treated with mycorrhizae, which effectively reduced pathogen impact. Disease recovery rates of up to 58% were achieved depending on the mycorrhizal type and pathogen involved. This significant reduction in pathogen severity can be attributed to the modulation of plant nutrient uptake, changes in root morphology, and competition between mycorrhizal fungi and pathogens for colonization sites, as demonstrated by Spagnoletti et al. (2021). Although differences in disease suppression rates among various mycorrhizal treatments were generally not significant, except for one pathogen group, the CE application was notably more effective, showing the highest suppression in three of the four pathogen groups. This superior performance of CE is attributed to its greater positive impact on root fresh and dry weights in infected plants.

Researchers testing different combinations of mycorrhizae against the root rot disease pathogen Rhizoctonia solani in watermelon found that AMF fungi helped to reduce the severity of the disease caused by soil-borne pathogens (Wu et al., 2021). Researchers obtained mix of AMF species (different species including FM and RI) and also a mix of AMF from different genera. Their findings were the opposite of those of our study. They observed that the blend of AMF from different genera performed exceptionally well in watermelon, showing notable improvements in terms of dry weight, photosynthesis rate, percent root colonization, and mycorrhizal dependence. Furthermore, the researchers reported that mycorrhizae from different genera exhibited greater efficacy compared to combinations of mycorrhizae from the same genus but different species, as well as individual mycorrhizae. Hafez et al. (2013) found that AMF mycorrhizae, including FM and RI, had significant effects on disease severity, disease suppression rate, and plant growth parameters in the bean plant when a mix mixture of AMFs was tested against the soil-borne pathogen R. solani. This study results demonstrated that the use of a single AMF species yielded better outcomes compared to a mixture of AMF species. This could be attributed to several factors, including competition between different AMF species for resources and space within the rhizosphere, which may reduce overall effectiveness. Additionally, specific plant-AMF compatibility likely plays a crucial role, where certain AMF species establish a more efficient symbiotic relationship with the plant. Moreover, a single AMF species may be more efficient in resource allocation and better adapted to the specific environmental conditions of our study. These findings suggest that, contrary to the expected synergistic interactions, a single AMF species might offer a more optimal solution for enhancing plant growth and health. The research by Demir et al. (2023) found that both individual treatments with AMF species FM and Gm, as well as their combined application, significantly reduced the severity of diseases caused by three major pathogens—Rhizoctonia fragariae, Fusarium oxysporum, and Alternaria alternata—in strawberry plants compared to control treatments. However, there was no significant difference in disease reduction between the combined AMF treatment and the individual AMF treatments. These results suggest that while AMF applications are effective in mitigating pathogen impact, combining FM and Gm does not provide additional benefits over using either species alone for controlling these specific soil-borne pathogens.

The performance of mycorrhizae in disease suppression may vary depending on the host variety, mycorrhizal variety and environmental conditions (Dowarah, Gill & Agarwala, 2022; Demir et al., 2023). FM reduced the disease severity of P. capsici by 57.2%, 43%, and 91.7% in field greenhouse and controlled climate room conditions, according to Ozgonen & Erkilic (2007). In the current study, FM AMF significantly reduced the severity of disease in 18-F. solani (41.66%), 50-F. mix (22.23%), and 48-F. solani (29.62%), but significantly increased the severity of disease in 147-F. mix (16.67%). These findings from our study and those of Ozgonen & Erkilic (2007) show that FM is effective against P. capsici and F. solani pathogens, but not against Fusarium mix disease pathogens.

Spagnoletti et al. (2021) demonstrated that inoculation with Rhizophagus intraradices (R. intraradices) significantly reduces disease severity and enhances plant growth parameters. Specifically, this treatment resulted in a 75.7% reduction in Fusarium pseudograminearum pathogen density and a 39% decrease in disease severity. These effects are attributed to increased antioxidant enzyme activity and decreased lipid peroxidation, which indicate improved redox balance and reduced oxidative stress in AMF-inoculated plants. Additionally, R. intraradices alleviates pathogen impact by competing for root colonization sites, thereby limiting pathogen establishment and enhancing plant tolerance.

Demir et al. (2023) observed that the application of AMF, including Funneliformis mosseae and Gigaspora margarita, significantly reduced the severity of soil-borne fungal diseases such as Rhizoctonia fragariae, Fusarium oxysporum, and Alternaria alternata in strawberry plants. Specifically, disease severity for Rhizoctonia fragariae decreased from 81.25% to 20.00%, for Fusarium oxysporum from 50.00% to 25.00%, and for Alternaria alternata from 42.50% to 20.00%, reflecting reductions of approximately 75%, 50%, and 53%, respectively. These reductions can be attributed to AMF’s impact on several underlying mechanisms. AMF improves nutrient uptake and enhances plant physiological functions, which contribute to higher levels of total phenolic content, antioxidant activity, and phosphorus content.

In the study conducted by Liu et al. (2018), it was observed that although AMF positively contributed to plant growth, it did not show any effect against powdery mildew in Standing milkvetch (Astragalus adsurgens), a legume forage plant. In fact, the disease index was higher in AMF-inoculated plants compared to plants that were not inoculated with AMF. Additionally, the researchers reported that there was no association between AMF colonization rate and crop morphological parameters or defense enzyme activities in the crop.

Conclusions

The study revealed that various AMF and a mycorrhizal mix have differing effects on pepper plants. The rates of disease severity suppression varied according to pathogen groups and mycorrhizae. Individual application of mycorrhizae was found to be more effective in suppressing disease severity than mixture application. Because of their higher positive impacts on plant root and vegetative parts, CE and RI have been shown to be more effective in disease suppression. While mycorrhiza mix was important in balancing the decreases in plant nutrient content following the application of the pathogen, it did not contribute to the correction of the decreases in plant growth parameters. Notably, the combination of MM with 50 F. mix pathogen was found to significantly enhance Ca, Fe, and K levels. In general, excluding FM, plants treated with mycorrhiza in conjunction with the pathogen colonized at a higher rate than plants treated with mycorrhiza alone. There is no clear result in terms of total number of spores, with some cases showing a decrease with the application of the disease and others showing an increase. However, plants treated only with mycorrhizal mix had the highest total spore count.

Supplemental Information

Supplemental Information 1 Colonization data.

Supplemental Information 2 Growth media data.

Supplemental Information 3 Number of spores.

Supplemental Information 4 Plant Nutrients.

Supplemental Information 5 Morphological data.

Supplemental Information 6 Disease severity 48Fsolani.

Supplemental Information 7 Disease severity 147Fmix.

Supplemental Information 8 Disease severity 50Fmix.

Supplemental Information 9 Disease severity 18Fsolani.

Supplemental Information 10 Two factor Anova table.

Supplemental Information 11 a) Biological control trial in the climate room of the GAPTAEM b) Roots of plants with Trypan Blue used to determine Mycorrhizal Fungi Spores Density c) A sample from the Root Colonization Count.

Supplemental Information 12 A view of AMF fungi (treatment from RI+ 18 F. solani) in the roots of pepper under the light microscope.

Supplemental Information 13 A view of AMF fungi (treatment from CE+ 48 F. solani) in the roots of pepper under the light microscope.

Supplemental Information 14 A view of AMF fungi (treatment from RI+ 147 F. mix) in the roots of pepper under the light microscope.

Supplemental Information 15 A view of AMF fungi (treatment from MM+ 50 F. mix) in the roots of pepper under the light microscope.

Supplemental Information 16 A view of AMF fungi (treatment from MM+ 147 F. mix) in the roots of pepper under the light microscope.

Supplemental Information 17 Pots used as Mycorrhizal Controls in the experiment (from left to right: Control Negative; FM; RI; CE; MM).

Supplemental Information 18 Pots in the 1st repetition used in 147-Fusarium mix applications in the experiment (from left to right: Control Negative; Control+ Pathogen; FM+P; RI+P; CE+P).

Supplemental Information 19 Comparison of pepper plants of each of the treatments compared for 147- Fusarium mix pathogen at the end of the experiment (ordered from left to right: Control N; Control P; FM+P; RI+P; CE+P; MM+P).

Supplemental Information 20 Comparing root structure of treatment 48-Fusarium solani+Mikoriza mix (left to right:MM control; MM+ 48 F. solani 1. Rep. ; MM+ 48 F. solani pathogen 2. Rep.; MM+ 48 F. solani 3. Rep.;MM control).

Supplemental Information 21 Comparison of pepper plants of each of the treatments compared for 18-F.solani pathogen at the end of the experiment (ordered from left to right: Control N; Control P; FM+P; RI+P; CE+P; MM+P).

Supplemental Information 22 The seedling of pepper showing pathogen disease severity in the experiment, with a severity scale of 4.

Supplemental Information 23 The disease symptoms caused by the pathogen.

Supplemental Information 24 The disease symptom wilt caused by the pathogen at the pathogenicity.

The author would like to express gratitude to Prof. Dr. Semra Demir (Van-Yüzüncüyil University, Faculty of Agriculture) for the AMF fungi she provided from her own work to be used in my studies and for the technical support; to Prof. Dr. Ali Volkan Bilgili and Dr. Hamza Yalcin (Sanliurfa-Harran University, Faculty of Agriculture) for statistical analysis; to Dr. Hatice Kara for analysis of plant nutrition; and to Hakan Bilgili for his help with graphic design, figure preparation, and in climate chamber and laboratory work for the article.

Additional Information and Declarations

Competing Interests

Author Contributions

Data Availability

The authors declare that they have no competing interests.

Ayşin Bilgili conceived and designed the experiments, performed the experiments, analyzed the data, prepared figures and/or tables, authored or reviewed drafts of the article, and approved the final draft.

The following information was supplied regarding data availability:

The raw data are available in the Supplemental Files.

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
