# Peer review of "The effectiveness of arbuscular mycorrhizal fungal species (Funneliformis mosseae, Rhizophagus intraradices, and Claroideoglomus etunicatum) in the biocontrol of root and crown rot pathogens, Fusarium solani and Fusarium mixture in pepper"

_PeerJ, doi:10.7717/peerj.18438_

## Round 0.1 · original submission · Major Revisions

Author has performed various experiments during this study. However, there are many flaws in this manuscript. Author has not selected an appropriate title. This study lacks novelty and all sections of the manuscript need improvement.

Please revise and resubmit

·

Basic reporting

no comment

Experimental design

no comment

Validity of the findings

no comment

Additional comments

Authors of the manuscript “The effectiveness of three Glomus species (G. mosseae, G. intraradices, and G. etunicatum) in the biocontrol of root and root rot pathogens Fusarium solani and Fusarium mix. in Pepper Plants” revealed that the effectiveness of AMFs varies depending on the specific strain of Fusarium solani, with better results observed against strains that cause less severe plant disease.
It is an interesting study. However, authors need to address the following issues:
1. Please add the implication of this research.
2. Please add data in results section of abstract.
3. Please add keywords in the manuscript. (And rearrange keywords in alphabetical order, available at online section).
4. Introduction: The introduction and background are reasonable. Some modifications must be incorporated.
• Why pepper plant? Please provide justification?
• Please delete unnecessary details from the introduction section.
• Authors have used very old references; it is better to replace them with the latest ones.
• References should be in chronological order (from older to recent).
• In line 45, what is this (--)? Please remove it.
5. Methodology: Subheadings should be bold.
• L 216, fig. 3.21? Please correct it.
• Please mention, one way or two way ANOVA?
• The Methodology section is very long. Delete unnecessary detail.
6. Results: Clearly explained, however extensively; reconsidering a more concise way of results description will provide a more comprehensive reading of this part, focusing only on significant effects of the treatments under investigation and excluding the non-significant effects. Also, attention should be given to avoid repetition in this part, please check the paragraphs started at L 287 and L297, both are almost same with similar wording.
• Please follow the same sequence of all parameters throughout the manuscript (Materials and methods, results and discussion section)
• Figures should be in High Resolution.
7. Discussion: There is a lack of mechanistic approach. Improve discussion with the help of logics and further relevant recent literature.
• L377, Please remove the word “discussion”.
• L416, start the sentence with a capital letter.
8. Conclusions: The contents of this section are appropriate.
9. References: Check the reference formatting manually, there are a lot of errors. Keep uniformity among references.
• Please don’t highlight any text e.g. see L520, L525 etc.
• Please add recent references.

Reviewer 2 ·

Basic reporting

This paper entitled “The effectiveness of three Glomus species (G. mosseae, G. intraradices, and G. etunicatum) in the biocontrol of root and root rot pathogens Fusarium solani and Fusarium mix. in Pepper Plants” studied the control effect of three AM fungi and mixed inoculation on pepper soil-borne diseases. The control effect is explained from the perspective of pepper biomass, Fusarium inhibition effect, and nutrient absorption, providing a foundational understanding for using AM fungi to manage soil-borne diseases. However, there are still some errors in the article that need to be clarified and corrected by the author:
According to the latest classification of Glomus, Glomus mosseae has been updated to Funneliformis mosseae, G. intraradices has been updated to Rhizophagus intraradices, and G. etunicatum has been updated to Claroideoglomus etunicatum (please refer to https://invam.ku.edu/species-descriptions). It is inappropriate for the author to use Glomus to summarize them. Please correct the whole manuscript.
In the introduction, the author elaborated on the role of pepper and the prevention and control methods of diseases, and introduces the role of AMF based on biological control. However, there was not mentioned whether such a situation exists in the region (GAP Region). Where the study was conducted? whether others have conducted similar research, the current progress of such research, and the importance of this study? Additionally, the cited references were relatively outdated. It is recommended that the author revise the introduction to explain why this study was conducted and its importance.
The references are generally outdated and lack novelty. Additionally, the format of the references are inconsistent, with some entries missing DOIs. Please review and revise them carefully.
The author uses various formats of numbers with "," and "." in the tables and includes lowercase letters, but does not indicate what these symbols represent. Additionally, according to the significant results, the correct number format should be the “mean ± standard error” of each indicator. Please correct these errors.
The figures in the article need to effectively illustrate the results related to the scientific question, particularly focusing on biomass and mycorrhizal colonization rates. It is essential to include clear and detailed figures of the mycorrhizal structure. The result images do not meet the requirements of journal publication at present.

Experimental design

The materials and methods section of the article lacks clarity. The experimental design, material processing, and statistical methods should be presented to readers in a clear and systematic manner.

Validity of the findings

In the results, the experiment involved the treatment of pepper with pathogens and grades different disease degrees. Therefore, detailed figures of the disease were necessary. In addtion, the author mentioned that "pathogens have a negative impact on morphological parameters" and "AM fungi can compensate for the effects of pathogens on morphological parameters," but only explains that the treatment has a significant impact on plant growth. The results did not clearly explain how much biomass and nutrients were reduced by disease treatment compared with the control, and how much of the loss caused by the disease is compensated by the inoculation of AM fungi. Please supplement the data in detail to better explain the results.
The first paragraph of the discussion is more like the purpose and significance of the study. It would be more appropriately introduced in the introduction but is somewhat redundant in the discussion. Please revise this section.

Additional comments

The English grammar contains some errors, and there are many long sentences in the article, such as in lines 99, 172, and 281. Please review the entire article and modify the word order appropriately to improve readability.

Annotated reviews are not available for download in order to protect the identity of reviewers who chose to remain anonymous.

Reviewer 3 ·

Basic reporting

I have gone through the MS entitled “The euffectiveness of three Glomus species (G. mosseae, G. intraradices, and G. etunicatum) in the biocontrol of root and root rot pathogens Fusarium solani and Fusarium mix. in Pepper Plants” Comments are as under;

1- The title needs to be revised as per PEERJ style. Moreover, full genus names need to be added at first mention not only in the title but in the whole MS.
2- Should be no full stop with in the title.
3- Expression of English needs to be improved like “In this research, the effectiveness of naturally derived arbuscular mycorrhizal fungi …..”
In fact, arbuscular mycorrhizal fungi are all natural.
4- Raw data not complete, each replicate value needed as raw data used for analysis. Disease raw data? Files need to be in English.
5- No need for figure 1, if available added figure showing
6- Figure 2 can be put as supplementary file.
7- Figure 4-5 reformat colors as clearly visible
8- Hypotheses not well stated.
9- Introduction lacks recent literature. The authors may consider revising it
10- Botanical name of pepper needs to be mentioned in title or in the abstract
At first mention.
11- In results, heading should be bold.
12- p>0.05 should be revisited.
13- In discussion section, Discussion appeared twice?
14- line 383-385 need to be revisited to delete old refences and add latest ones.
15- cross math the references also format to match PEERJ style
16-Table 3, need to be consulted with a statistician. e.g., abc be written as a-c. There looks errors in lettering also. All tables and data need to be consulted with statistician to reanalyze.
Analysis done column wise or row wise need to be mentioned.

17- All tables should be self explanatory, abbreviations description need to be added in each and every table

18-All scientific names should be italics in the whole MS.

Experimental design

1- The MS falls within the scope of the journal
2- Research question not well defined. Novelty statement is obscure.
3- There should be bold headings in methods
4- line 215 “mycorrhizal spores in the roots” (authors need to provide reference/evidence that spores are produced with in plant roots. Or outside plant roots? Some mycorrhizal species don’t produce spores in side roots. Evidences in the form of references required about the mycorrhizal species used in the present study.
5- In methods, what were the spore counts of pathogens used?

Validity of the findings

1- Impact and novelty not assessed.
2- The composition of Fusarium mix not clear.
3- In abstract “Single AMF species were more effective in enhancing the growth of pathogen-treated host plants and suppressing disease compared to the mixed AMF”.
& “The mixed AMF was only more eûective in balancing pathogen-induced decreases in plant nutrients (Cu, Mg, Zn, P).”
Authors need to address the possible reasons behind this effect in discussion.
4- conclusion needs to be rewritten, avoiding methodology.
5- Raw data not complete.

---

## Round 0.2 · Minor Revisions

Dear Authors,
Please improve your manuscript by incorporating the corrections suggested by the reviewers.

·

Basic reporting

No comment

Experimental design

No comment

Validity of the findings

No comment

Additional comments

Please modify the discussion section.

Please elaborate on the underlying mechanisms driving the observed changes in parameters and add references for that mechanism.

For example, ''The significant alteration in parameter X can be attributed to the modulation of [specific biological pathway], as demonstrated by [Reference 1: Reference2].''
Ensure that the discussion is evidence-based and thoroughly referenced, incorporating recent findings and scholarly perspectives to validate the argument.

Reviewer 2 ·

Basic reporting

Thanks to the author for patiently revising the manuscript according to the comments. While the quality of the article has improved, there are still issues that need further attention:

1. Abstract: The introduction in the abstract is unclear. The study only involved two pathogens, yet the results suggest that the effectiveness of AMF in controlling pepper diseases varies depending on the pathogen. The experimental results need to be explicitly communicated to the readers.

Experimental design

2. Introduction: The author has revised the introduction, but some sections remain too lengthy. For example, in line 57, it is mentioned that these diseases pose a significant threat to pepper production, followed by a description of pepper root rot symptoms in line 62. It is recommended that the author condense these points and summarize the content more concisely.

Validity of the findings

3. Data analysis: The treatment used by the author is AMF + pathogens, which is a two-factor treatment, but the single-factor analysis written in this part is obviously incorrect. Therefore, it is not obvious in the conclusion whether the treatment combination of AMF and pathogen is more significant than others. Please analyze which specific treatment combination of AMF and pathogen is the most significant under the two-factor treatment, and modify it in the abstract and conclusion.

Additional comments

4. There are still some minor errors in the author's article that have not been corrected, such as in line 260, NAClO is mistakenly written as NAOCl. There is also a missing period at the end of the data analysis paragraph. Please check the article carefully and make revisions.

Annotated reviews are not available for download in order to protect the identity of reviewers who chose to remain anonymous.

Reviewer 3 ·

Basic reporting

I have gone through the MS entitled “The effectiveness of three Arbuscular Mycorrhizal Fungi species (Funneliformis mosseae, Rhizophagus intraradices, and Claroideoglomus etunicatum) in the biocontrol of root and crown rot pathogens Fusarium solani and Fusarium mixture in Pepper” Comments are as under;

1- The title needs to be revised as ‘The effectiveness of arbuscular mycorrhizal fungal species (Funneliformis mosseae, Rhizophagus intraradices, and Claroideoglomus etunicatum) in
the biocontrol of root and crown rot pathogens, Fusarium solani and Fusarium mixture in Pepper’
2- Figure 1, put as supplementary
3- Figure 4- 5 quality should be improved

Experimental design

1- The MS falls within the scope of the journal

2- The authors did not address my previous comment “mycorrhizal spores in the roots” (authors need to provide reference/evidence that spores are produced with in plant roots. Or outside plant roots? Some mycorrhizal species don’t produce spores in side roots. Evidences in the form of references required about the mycorrhizal species used in the present study.
Put reference proving the mycorrhizal species used in the present study produce spores inside the root.

Validity of the findings

Put reference proving the mycorrhizal species used in the present study produce spores inside the root.

---

## Round 0.3 · Minor Revisions

Please address reviewers comments. The microscopic pictures do not show spores as these may be arbuscules or vesicles? not spores. as pics are not of high resolution. This can not be considered as evidence. Just send 1-2 previous studies proving that these mycorrhizal species produce spores in side roots.

·

Basic reporting

No comment

Experimental design

No comment

Validity of the findings

No comment

Additional comments

Thank you for revising the discussion section, it is now clearer and more comprehensive. However, I noticed that the scientific names mentioned in the text are not consistently formatted in italics. Please ensure that all scientific names are formatted correctly throughout the manuscript.

Reviewer 3 ·

Basic reporting

The authors have revised the mS but still formatting issues are there.

Experimental design

-

Validity of the findings

All bars in figures dont have y error bars? Authors need to address carefully.
Authors need to show some previous study claiming that the mycorrhizal species used in the present study produce spores in side roots. The microscopic pictures does not show spores as these may be arbuscules or vesicles? not spores. as pics are not of high resolution. This can not be considered as evidence. Just send 1-2 previous studies proving that these mycorrhizal species produce spores in side roots.

---

## Round 0.4 · accepted · Accept

Authors have addressed all of the reviewers comments. This manuscript is ready for publication.